# Directing polymorph specific calcium carbonate formation with de novo protein templates

Fatima A. Davila-Hernandez[1,2,3,8], Biao Jin[4,5,8] ✉, Harley Pyles[1,2,8], Shuai Zhang [4,5], Zheming Wang[4], Timothy F. Huddy[1,2], Asim K. Bera [1,2], Alex Kang[1,2], Chun-Long Chen [4,6], James J. De Yoreo [4,5] ✉ & David Baker [1,2,7] ✉

Biomolecules modulate inorganic crystallization to generate hierarchically structured biominerals, but the atomic structure of the organic-inorganic interfaces that regulate mineralization remain largely unknown. We hypothesized that heterogeneous nucleation of calcium carbonate could be achieved by a structured flat molecular template that pre-organizes calcium ions on its surface. To test this hypothesis, we design helical repeat proteins (DHRs) displaying regularly spaced carboxylate arrays on their surfaces and find that both protein monomers and protein-$Ca^{2+}$ supramolecular assemblies directly nucleate nano-calcite with non-natural {110} or {202} faces while vaterite, which forms first in the absence of the proteins, is bypassed. These protein-stabilized nanocrystals then assemble by oriented attachment into calcite mesocrystals. We find further that nanocrystal size and polymorph can be tuned by varying the length and surface chemistry of the designed protein templates. Thus, biomineralization can be programmed using de novo protein design, providing a route to next-generation hybrid materials.

Organisms generate biominerals such as nacre, shell, bone, and tooth[1,2] by controlling the nucleation, growth, and assembly of inorganic crystals[3–7]. The sequences of native biomineral-associated proteins and their effects on mineralization are well characterized for some systems[4,8–16] but the three-dimensional structures of the protein-mineral interfaces are largely unknown. For example, natural proteins nucleate specific calcium carbonate ($CaCO_3$) polymorphs[10,11,17–19], but none have known stable tertiary structures and many are intrinsically disordered or insoluble[20–22]. Additives including small organic molecules[23–25], polymers[26], amino acids[27], and peptides[28,29] affect $CaCO_3$ crystallization but have not been shown to template nucleation.

$CaCO_3$ growth modification and localization of nucleation have been explored with short unstructured $CaCO_3$ binding peptides genetically fused to proteins immobilized on a surface and with post-translational modified proteins[30,31], but in such systems, it is difficult to structurally define the protein-mineral interface. Stereochemical-matched self-assembled monolayers (SAMs)[32] modulate the nucleation of $CaCO_3$, suggesting that geometric lattice matching may provide a general route to controlling mineralization[3,4,33,34]. In further support of this concept, the structures of ice-binding proteins contain surfaces with an epitaxial-like lattice matching the ice lattice[35], which enables modulation of ice formation by binding ice nuclei through preorganized

[1]Department of Biochemistry, University of Washington, Seattle, WA 98195, USA. [2]Institute for Protein Design, University of Washington, Seattle, WA 98105, USA. [3]Molecular Engineering Graduate Program, University of Washington, Seattle, WA 98105, USA. [4]Physical Sciences Division, Pacific Northwest National Laboratory, Richland, WA 99352, USA. [5]Department of Materials Science and Engineering, University of Washington, Seattle, WA 98195, USA. [6]Department of Chemical Engineering, University of Washington, Seattle, WA 98195, USA. [7]Howard Hughes Medical Institute, University of Washington, Seattle, WA 98105, USA. [8]These authors contributed equally: Fatima A. Davila-Hernandez, Biao Jin, Harley Pyles. ✉e-mail: biao.jin@pnnl.gov; James.DeYoreo@pnnl.gov; dabaker@uw.edu

ice-like waters, and ferritin has been shown to organize iron clusters on its interior surface[36]. Advances in protein design now enable the creation of new proteins with precisely specified structures, and in previous work, we showed that lattice-matched interactions between arrays of carboxylate side chains on a designed protein surface and pre-formed mica crystals direct formation of precise protein–mica assemblies[37].

Guided by the ice-binding protein example, we hypothesized that designed proteins with flat surfaces displaying functional groups lattice-matched to a mineral of interest could be used to modulate mineralization by promoting ion association in a pattern consistent with the mineral lattice, thus directly reducing the free energy of formation of the critical nucleus (Fig. 1a). As a first test of the potential of designed proteins for directly nucleating inorganic mineralization, we chose $CaCO_3$ as a model system due to its numerous polymorphs, well-documented nucleation behavior with and without proteins[38], and prevalence in natural biominerals, and sought to design proteins with structures suitable for nucleating crystalline $CaCO_3$ (Fig. 1a). We aimed to generate flat repetitive surface arrays of calcium-coordinating carboxylate groups (Fig. 1b) that could bind to and stabilize $CaCO_3$ nuclei (Fig. 1c). To enable exploration of the contribution of the size of the protein-mineral interface while keeping the chemical composition and geometry otherwise constant, we employed Designed Helical Repeat (DHR)[39] proteins that can be shrunk or extended simply by changing the number of repeating units. These proteins have >10 nm² flat surfaces with greater structural definition and tunability than SAMs displayed on genetically encoded soluble molecules that can be readily reprogrammed.

Here, we describe the design of DHR proteins that template growth of $CaCO_3$ crystals. We show these proteins promote the direct nucleation of calcite nanoparticles under conditions which otherwise yield vaterite microcrystals and change the post-nucleation growth from a classical pathway to orientated particle attachment. These results demonstrate the power of designed proteins for guiding the formation of hybrid materials.

## Results

### Design and characterization of DHR proteins
Flat repeat helix-turn-helix-turn protein backbones were designed with either constrained monte-carlo fragment assembly with RosettaRemodel[40], or by placing α-helices in a plane and then connecting them with additional secondary structural elements[41]. In contrast to typically designed protein surfaces containing a mix of positive, negative, and non-charged hydrophilic residues to promote solubility, and to hydrophobic protein surfaces designed to bind other proteins, we designed surfaces consisting entirely of carboxylate side chains spaced at regular intervals to mimic the carbonate sublattice on different crystallographic planes of $CaCO_3$ crystals (see "Methods").

We selected 40 designs for experimental characterization based on criteria described in the methods section and obtained synthetic genes encoding them (Supplementary Table 1). Eight of these expressed at sufficiently high levels in *E. coli* and were highly soluble. Two designs, FD15 and FD31, which were monodisperse in size exclusion chromatography experiments (SEC; Supplementary Fig. 1a, c) were selected for detailed characterization. Circular dichroism (CD) experiments showed the expected alpha-helical structure, and the proteins were found to be hyperstable, remaining folded at temperatures up to 95 °C (Fig. 1e, h). Small-angle X-ray scattering (SAXS) profiles of FD31 were consistent with profiles computed for the design model (Fig. 1i). The crystal structure of FD15 was solved at 4 Å resolution and matched the design model with an RMSD of 0.45 Å (Fig. 1f) with a perfectly straight repeating structure having a 8.7 Å spacing between helices in adjacent repeat subunits. To allow for protein–protein assembly, as in some native biomineralization proteins[16,42], we also designed an array of carboxylate side chains onto the surface of DHR49, a previously designed flat DHR[39], and removed the capping elements on the terminal repeats to produce head-to-tail protein-protein interfaces. This protein (DHR49-Neg) was alpha-helical and thermostable (Supplementary Fig. 1d) and atomic force microscopy (AFM) experiments showed it indeed assembled end-to-end to form single-protein diameter fibers at the mica–water interface (Supplementary Fig. 1e).

### Control of $CaCO_3$ nucleation by designed proteins
We first investigated the effect of the designed proteins on $CaCO_3$ mineralization using in situ liquid-phase attenuated total reflectance-Fourier transform infrared spectroscopy (ATR-FTIR) and time-dependent ex situ transmission electron microscopy (TEM) (Fig. 2). We added 1.08 μM of each designed protein or a BSA control to a mineralization solution containing 5 mM $CaCl_2$ and 5 mM $NaHCO_3$. At

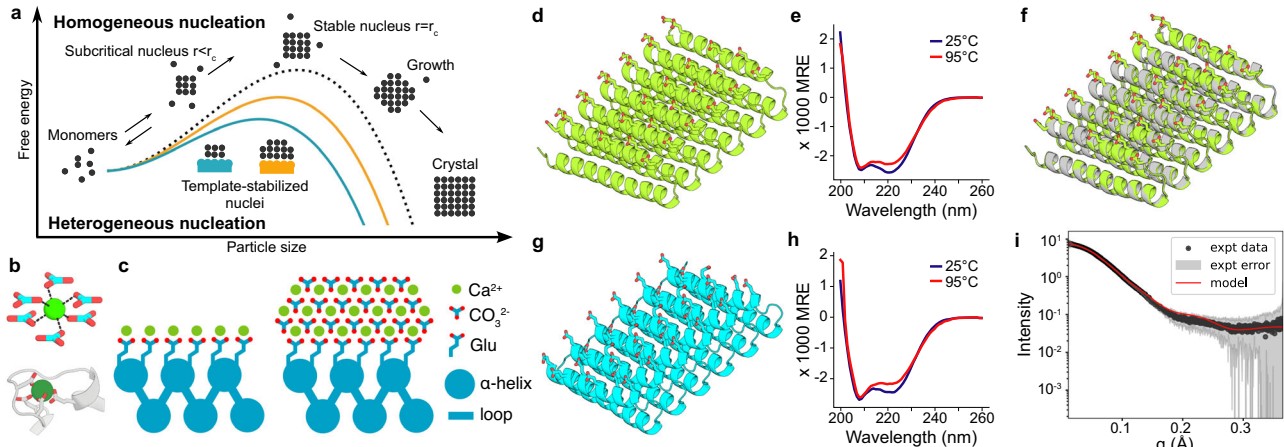

**Fig. 1 | Design principles for the heterogeneous nucleation of $CaCO_3$ guided by a protein template. a** The presence of a template reduces the free energy barrier of nucleation and the critical radius of the nuclei. **b** Coordination of calcium (top) by carbonate ions within a unit cell of calcite, and (bottom) by carboxylate containing glutamate and aspartate residues in the structure of calmodulin (PDB ID: 1A29, residues 18–33). **c** Tessellating binding moieties across repeated α-helices within a designed protein capable of pre-organizing bound calcium ions. **d–i** Biophysical characterization of the designed proteins. Design models for (**d**) FD15 and (**g**) FD31. **e, h** Their respective circular dichroism scans showing mean residue ellipticity (M.R.E.) from 200 to 260 nm at 25 °C and 95 °C. **f** The 4 Å resolution crystal structure of FD15 in gray (pdb id: 8UGC) superimposed on the design model in green (RMSD of 0.45 Å). **i** Experimental and model SAXS profiles for FD31, scattering vector (q, from 0 to 0.35 Å⁻¹) vs. intensity.

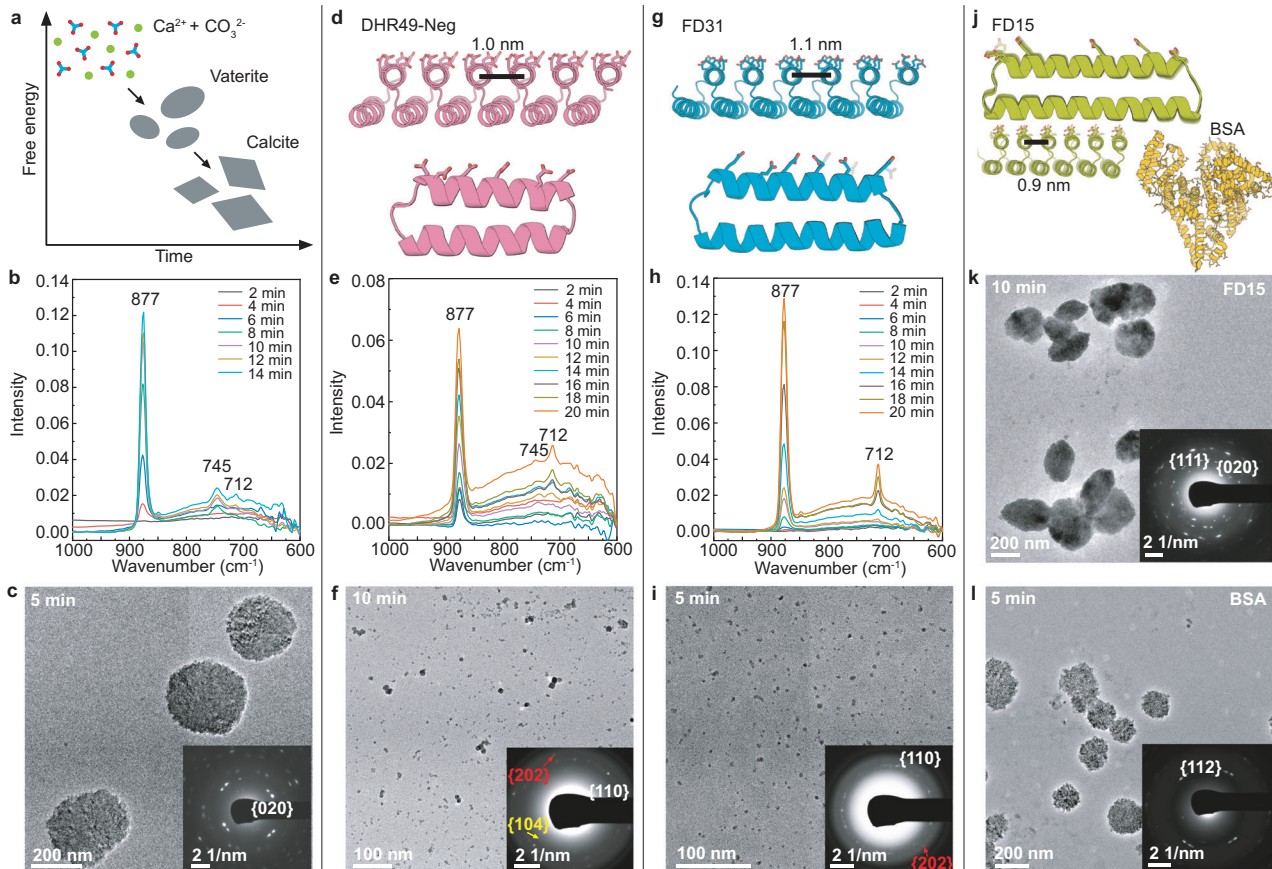

**Fig. 2 | Designed proteins modulate CaCO₃ crystallization.** The CaCO₃ crystallization process in supersaturated solutions containing 5 mM CaCl₂ and 5 mM NaHCO₃ was monitored in the absence and presence of different designed proteins. **a** Schematic showing the nucleation and transformation of CaCO₃ in the absence of additives under the conditions used here. First, vaterite forms followed by transformation to micron-sized calcite[43]. **b** In situ ATR-FTIR and **c** TEM of CaCO₃ crystallization in the absence of protein showing the formation of vaterite. **d**–**l** Impact of 1.08 μM designed proteins on mineralization. Protein design models (**d**, **g**) and corresponding in situ ATR-FTIR (**e**, **h**) show the formation of predominantly calcite in the presence of DHR49-Neg, and exclusively calcite in the presence of FD31. Representative TEM images confirm the formation of (**c**) vaterite in the absence of the proteins, **f** primarily calcite crystals with DHR49-Neg, and **i** calcite nanocrystals in the presence of FD31. In contrast, in the presence of (**j**) FD15 or a BSA control (**j**) only vaterite is formed (**k**, **l**). The TEM experiments in (**c**, **f**, **i**, **k**, **l**) were independently repeated three times with similar results.

this concentration, the solution is supersaturated with respect to all three common crystal phases—vaterite, aragonite, and calcite—but is undersaturated with respect to amorphous CaCO₃ (ACC), and the least stable phase, vaterite, is the first to nucleate in bulk solution[43]. Thus, we are able to investigate the ability of the protein templates to nucleate specific polymorphs and alter the nucleation pathway. Vaterite indeed appeared as the first solid phase in the protein-free mineralization solution, as well as in the presence of bovine serum albumin (BSA), which has the large negative charge of the DHR proteins but lacks both the periodic array of carboxylates and flat interface (Fig. 2b, c, l); the vaterite crystals grew to become hundreds of nm diameter spheroidal particles. In contrast, in the presence of DHR49-Neg predominantly calcite nanocrystals ≈ 5 nm in size were initially observed (Fig. 2e, f). FD31 was also highly effective at promoting the direct nucleation of nanocrystalline calcite (Fig. 2h, i) as indicated by selected area electron diffraction (SAED) (Supplementary Fig. 2a), cryo-TEM (Supplementary Fig. 2b), high-resolution (HR) TEM (Supplementary Fig. 2c), and liquid-phase (LP)-TEM images (Supplementary Fig. 2d). In contrast, FD15 did not alter the crystallization pathway from that seen with the no protein and BSA controls (Fig. 2k). Given that FD15, FD31, and DHR49-Neg all have flat surfaces with regularly arrayed carboxylate groups, these differences likely stem from differences in repeat spacing (Supplementary Fig. 3) and detailed surface chemistry (Supplementary Table 2).

## Revealing the nucleation mechanism of calcite nanocrystals

To investigate the mechanism underlying these nucleation effects, we focused on FD31, which effectively promoted direct formation of nano-calcite. Supersaturated solutions containing 1.08 μM protein were sealed into a liquid-cell for LP-TEM observations. Two nucleation pathways were observed. In the first, calcite nanocrystals nucleated throughout the solution (Fig. 3a and Supplementary Movie 1), reaching a diameter of ≈ 7 nm and exhibiting a number density comparable to that of the protein monomers ($\approx 1.1 \times 10^{15}$ vs $\approx 6.5 \times 10^{14}$; see Supplementary Discussion). Increasing the protein concentration from 0.5 μM to 1 μM and 2 μM led to an increase in the number density of nano-calcite particles (Supplementary Fig. 4a–c). Due to their low contrast and small size, individual protein monomers are not observable with LP-TEM, however, AFM indicates their presence in solution (Supplementary Fig. 5e). The calcite nanoparticles were also observed when imaging was performed at lower magnification and were already present when the electron beam was moved to neighboring regions (Fig. 3b), indicating that electron beam effects were not driving the process (see supporting text for more details). FD31 is 8 nm by 5 nm by 3 nm in size, which is similar to the 1–5 nm critical nucleus size of CaCO₃[44,45]. The similarity in size and number density of the calcite nanocrystals and protein monomers suggests nucleation is driven by individual proteins.

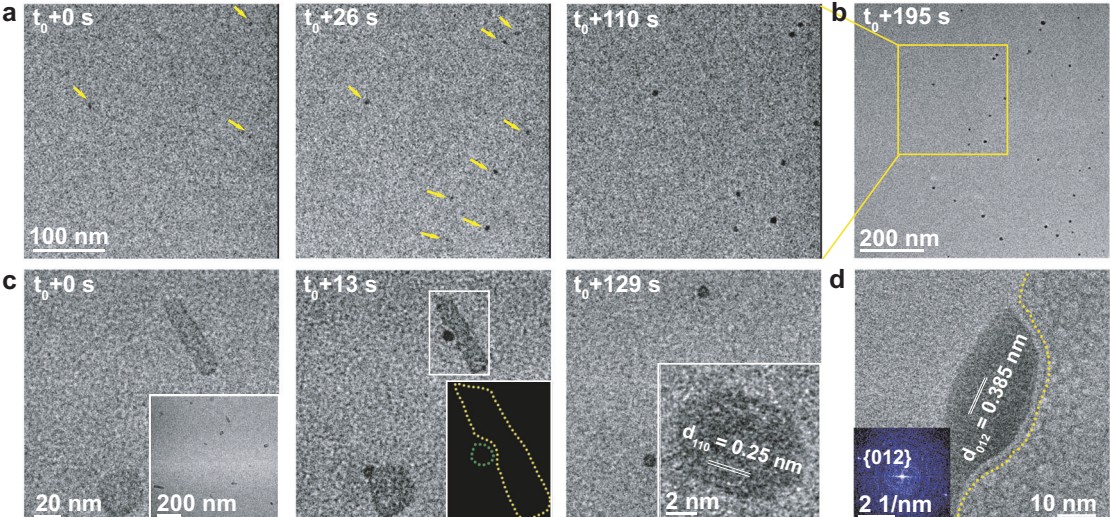

**Fig. 3 | FD31 mediated nucleation of calcite nanocrystals. a** Sequential LP-TEM images show the nucleation of eight CaCO₃ nanoparticles (marked by yellow arrows) in a precursor solution containing FD31 protein. **b** Low-mag LP-TEM image shows more nucleated nanoparticles. **c** Sequential LP-TEM images show the initial nucleation and growth of two CaCO₃ nanoparticles around the sheet-like templates after FD31 was preincubated with CaCl₂ for 5 min before the addition of NaHCO₃. Inset at $t_0 + 0$ s displays multiple template formations at an early stage. A schematic (Inset at $t_0 + 13$ s) shows the disconnected interface with a width of ≈ 1 nm between the as-formed particle and the substrate. At $t_0 + 129$ s, the protein assembly dissolves, likely due to depletion of Ca, and the protein disperses into solution. Inset at $t_0 + 129$ s shows an LP-HR-TEM image of a newly formed particle with the {110} lattice spacing of calcite. **d** In situ LP-HR-TEM image reveals the interface structure between calcite and the template. The yellow dotted lines highlight the interface. The FFT image in the inset shows the nucleated particle is calcite. As in Fig. 2g–i, the concentration of FD31 protein is 1.08 μM and CaCl₂ and NaHCO₃ is 5 mM. The TEM experiments were independently repeated three times with similar results.

We observed a second nucleation pathway driven by sheet-like assemblies when FD31 was incubated with Ca²⁺ prior to the addition of NaHCO₃ (Fig. 3c and Supplementary Movie 2). Pre-incubation of 2.16 μM FD31 with 10 mM Ca²⁺ in the absence of HCO₃⁻ led to the formation of supramolecular assemblies (inset in Fig. 3c). Upon addition of an equal volume of 10 mM NaHCO₃, after 5 min we observed the nucleation of nanoparticles adjacent to the protein-Ca²⁺ supramolecular assemblies (Fig. 3c, Supplementary Movie 2, and Supplementary Fig. 5j) identified as calcite by in situ high-resolution (HR)-TEM (Inset in Fig. 3c). As the calcite particles continued to grow, the protein-Ca²⁺ assemblies dissolved (Fig. 3c). Incubation of FD31 with CaCl₂ for 30 min to 24 h led to the coordination of Ca²⁺ by carboxylate groups (Supplementary Fig. 5a, b) accompanied by the formation of the protein-Ca²⁺ assemblies, which were found to be crystalline (Supplementary Figs. 5d–i and 6). The protein-Ca²⁺ assemblies likely dissolve as calcite particles grow because the available Ca²⁺ in their vicinity is depleted. Thus, while these assemblies are individually more effective templates than are the protein monomers, as demonstrated by the rapidity with which nucleation occurs in their vicinity, by the time the nucleation phase has ended, the fraction of the crystals created by the monomers far exceeds that produced by these assemblies.

At favorable viewing angles, a ≈ 1 nm separation was observed between the protein-Ca²⁺ assemblies and the nascent calcite nanocrystals (Fig. 3c, d). This is consistent with the ≈ 1 nm thickness of hydration layering on calcite[45] and suggests nucleation is an interface-driven process, as observed in other mineral systems in which surface ligands present carboxyl-rich interfaces[46,47]. The results suggest that the flat periodic array of acidic residues on the surface of the proteins directs the absorption and organization of Ca²⁺ ions and water molecules in the hydration layer into an arrangement consistent with the structure of calcite. Alternatively, the interfacial region may alter the activities of the CaCO₃ phases to modulate the supersaturation with respect to calcite vs vaterite.

## Determining the growth mechanism of calcite crystals

We next followed the growth of FD31 nucleated calcite nanocrystals over time using LP-TEM. During the ≈ 5 min following nucleation, individual nuclei assembled into 10–20 nm particles through multiple attachment events, rather than individually growing in size (Fig. 4a and Supplementary Movie 3). Ex situ HR-TEM showed that the larger agglomerated particles are surrounded by smaller particles (Fig. 4b, c), and the assemblies have continuous crystal lattices (Fig. 4d). Similar agglomerates were also observed after growth with DHR49-Neg at this time point (Supplementary Fig. 7). Based on these observations we conclude that FD31 and DHR49-Neg stabilize calcite crystals at ≈ 5 nm size and that oriented particle attachment[5] is the dominant pathway for calcite growth in their presence.

The ≈ 20 nm agglomerated calcite crystals (Fig. 4b) formed with FD31 and DHR49-Neg have different preferred facet orientations, {110} for FD31 (Fig. 4c–d) and {202} for DHR49-Neg (Supplementary Fig. 7c–f), both of which are distinct from the natural {104} facets of calcite grown in protein-free solutions. The differences between facet orientations for the two proteins likely reflect differences in their structure and chemistry (highlighted in Supplementary Table 2). First, while both are composed of repeating alpha helices, these are spaced at different distances, potentially leading to different extents of epitaxially matching for different calcite surfaces (Supplementary Fig. 3, Supplementary Discussion). Second, although the two proteins have similar net-charges and number of carboxylate groups, they contain different ratios of aspartate and glutamate residues, which present carboxylate groups in different orientations and may result in distinct stereochemical alignments relative to carbonate ions in the calcite surface (Supplementary Table 2 and Supplementary Discussion).

At 20 min in FD31 solutions, we observed ≈ 100 nm calcite crystals with a pseudo-rhombohedral morphology (Fig. 4e), their habit resembling the thermodynamically stable {104} calcite rhombohedron, but with terraced corners and low contrast features that suggest a discontinuous lattice separated by voids, perhaps created by the inclusion of proteins. At 30–40 min, the calcite crystals grew to

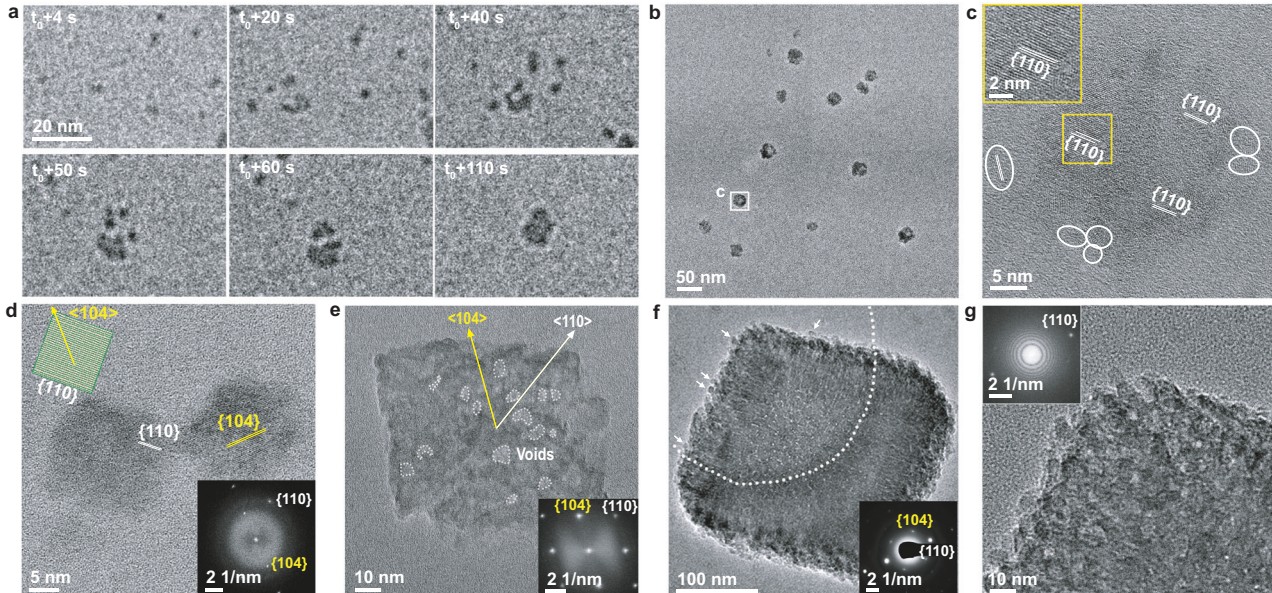

**Fig. 4 | Calcite nanocrystal growth and assembly in the presence of FD31.**
**a** Sequential LP-TEM image sequence shows the attachment of multiple calcite nanocrystals into a larger one. **b** Ex situ TEM image showing calcite grown through particle attachment. The mean measured particle size of the 12 particles shown is 22.8 ± 3.9 nm. **c** HR-TEM image showing the aggregated crystal with {110} lattice orientation. White frames mark several individual calcite nanocrystals around the larger particle. **d** HR-TEM image shows additional calcite nanocrystals with a size of ≈ 20 nm and {110} side faces. The inset shows the calcite morphology and facet information. **e** TEM image shows ≈ 100 nm rhombohedral calcite. The inset FTT shows the crystal orientation and single crystallinity. Yellow and white arrows represent <104> and <110> growth orientations, respectively. **f** TEM image showing rhombohedral calcite with a size of ≈ 300 nm. The embedded SAED image demonstrates its single crystalline nature. Arrows mark several calcite nanocrystals on the surface. **g** HR-TEM and corresponding FFT images in the inset confirm the single crystalline nature and that the rough surface is composed of attached nanoparticles. Similar TEM results were collected in five independent experiments.

~300–800 nm (Fig. 4f, g and Supplementary Fig. 8a, b), and retained a similar morphology with rough surfaces, internal voids, and multiple 4–8 nm particles on their surfaces (Supplementary Fig. 8c–h), suggesting growth through repeated attachment of the nanocrystals. This conclusion is supported by LP-TEM data (Supplementary Fig. 9). When the FD31 concentration was increased fourfold, the calcite crystals were elongated along the c-axis (<001> direction) (Supplementary Fig. 10). DHR49-Neg also formed calcite rhombohedral of similar size and habit (Supplementary Fig. 7g). Thus, in the presence of the proteins, the 4–8 nm primary calcite nanocrystals assemble through oriented attachment to ultimately form micron-scale mesocrystalline rhombohedral calcite.

**Exploring the effects of template size and surface chemistry**
We next investigated the features of FD31 that contribute to its potent calcite nucleation activity by generating versions of the protein with varied structural and chemical properties. The effect of template length (the extent of the carboxylate arrays) was investigated by generating versions of the protein with different numbers of repeat subunits. FD31 has 6 repeat units and a 5 × 8 nm interface containing 36 carboxylate groups. We produced a 3-repeat version with a 5 × 5 nm interface containing 18 carboxylate groups and a 9-repeat version with a 5 × 11 nm interface containing 54 carboxylate groups. The 3, 6, and 9-repeat templates all drive the formation of nano-calcite, but the 3-repeat version leads to larger nanoparticles (Fig. 5a–c), likely due to reduced ability to lower the interfacial energy or weaker binding to calcite surfaces following nucleation. Nucleation driven by monomers of FD31-Rep9 was also observed by liquid-phase TEM (Supplementary Fig. 11).

To investigate the effect of surface chemistry on calcite nucleation activity, we modified the surface chemistry of FD31 in three ways. We either substituted half of the glutamate residues to nearly isosteric but non-charged glutamine residues (FD31-Gln-Checker), replaced 24 out of 36 glutamate residues with shorter aspartic acid residues (FD31-

Asp), or mutated 13 out of 36 of the glutamate to basic lysine residues (FD31-Lys-Checker) (Fig. 5d, f, h). FD31 (containing an all-glutamate interface) and FD31-Gln-Checker generated calcite-dominant nanocrystals (Figs. 2g–i and 5e and Supplementary Fig. 12a, b), both vaterite and calcite phases appeared in the presence of FD31-Asp (Fig. 5g and Supplementary Fig. 12c, d), and FD31-Lys-Checker produced primarily vaterite (Fig. 5i and Supplementary Fig. 12e, f). Thus, the structure and chemistry of the sidechain arrays in our designed protein directly influence the templating of calcite nucleation.

## Discussion
Our designed protein templates promote the direct formation of 5–9 nm calcite nanocrystals through both monomer- and protein-Ca²⁺ assembly-driven nucleation, bypassing the formation of vaterite microcrystals that occurs in their absence or with other negative design controls. The protein assembly-driven pathway resembles the multistep process in bone and dentin calcification where Ca²⁺-induced assembly of an acidic matrix protein precedes the formation of apatite[48]. The proteins then inhibit further growth of the crystals directly from the solution, stabilize non-natural faces, and switch the crystallization pathway to nonclassical oriented attachment, ultimately producing micron-scale rhombohedral single crystals with terraced edges, abundant low contrast inclusions, and morphologies that vary with protein design and concentration. The direct appearance of calcite strongly suggests that the distributions of carboxylic side chains bias the configuration of the ions at the protein-solution interface towards that of the calcite lattice; the rigid structure of the designed proteins now provides an unparalleled opportunity to determine the detailed mechanism of control (see Supplemental Discussion).

Our DHR proteins possess greater structural regularity and tunability than previously studied native and engineered biomineralization proteins[11,31,49] and drive polymorph-specific CaCO₃ nucleation. This tunability allows rigorous interrogation of the interface driving this templating effect. We found that the spacing of the designed

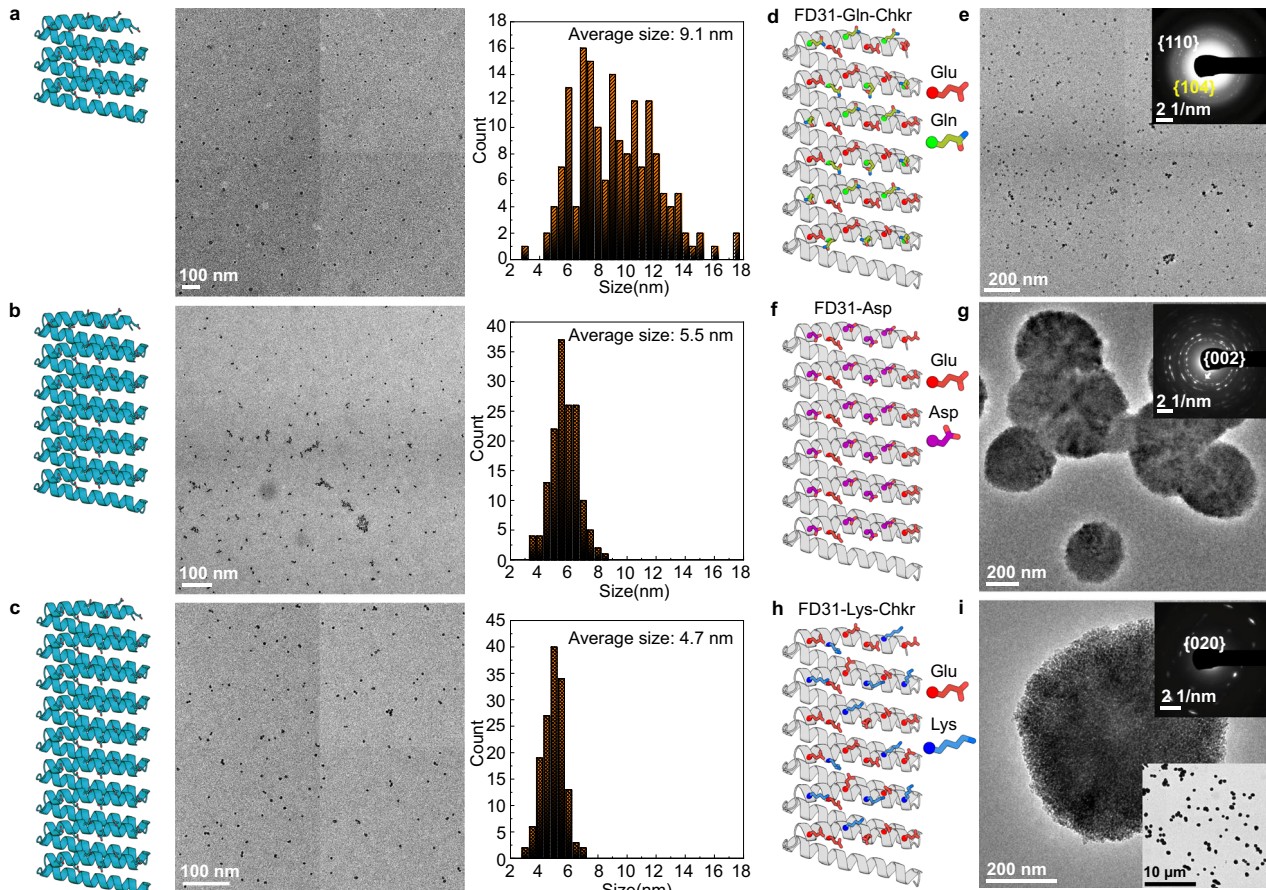

**Fig. 5 | Tuning calcite nucleation by varying FD31 length and surface chemistry.** **a**–**c** TEM and the size distribution of particles formed in the presence of **a** FD31-Rep3 with 3 repeats, **b** FD31 with 6 repeats, and **c** FD31-Rep9 with 9 repeats. The corresponding particle number (*n*) and mean measured particle size (*S*) are: **a** $n = 172$, $S = 9.1 \pm 2.9$ nm, **b** $n = 150$, $S = 5.5 \pm 0.9$ nm, **c** $n = 147$, $S = 4.7 \pm 0.8$ nm, respectively. All the measurements are based on the representative TEM images from the same sample. **d**–**i** Representative TEM and SAED images show the formation of (**e**) calcite-dominant nanocrystals in the presence of FD31-Gln-Checker (**d**); the formation of (**g**) vaterite as well as calcite (Supplementary Fig. 12c, d) in the presence of (**f**) FD31-Asp; and the formation of a (**i**) vaterite dominant phase in the presence of (**h**) FD31-Lys-Checker. Three independent TEM experiments were performed with similar results.

sidechain arrays with similar chemistry had a pronounced effect (Supplementary Table 2): FD15 with an array spacing of 0.9 nm did not template calcite, DHR49-Neg with a spacing of 1.0 nm templated calcite to some extent, and FD31 with a 1.1 nm spacing was a strong nucleator of calcite. The 1.0 nm and 1.1 nm spacings produced nanocrystals with different preferred orientations, potentially due to the formation of epitaxial matches with different calcite planes (Supplementary Figs. 3, 13, 14 and Supplemental Discussion). We also observed effects of the overall array dimensions: comparing 5, 8, and 11 nm long versions of FD31 (FD31-Rep3, FD31, FD31-Rep9, respectively) showed that the smaller templates nucleated larger calcite particles, either because they have higher interfacial free energies and are less effective nucleators, leaving more $Ca^{2+}$ and $CO_3^{2-}$ available for post-nucleation classical growth, or because they bind less tightly to the nucleated crystals and hence retard growth less (there is a linear scaling between inverse particle surface area and protein surface area (Supplementary Fig. 15). Given that template-mineral binding is directly related to the interfacial free energy between the template and the crystal[50], these two effects are expected to go hand in hand[51]. The ability to change the surface chemistry simply by recoding the synthetic gene encoding the designs enabled interrogation of the effect of surface chemistry on nucleation outcomes. The observation that FD31-Gln-Checker nucleates calcite better than FD31-Asp suggests that stereochemical alignment is more important to the calcite-promoting activity than the total number of negative moieties. Steric hindrance and/or the inclusion of

positive moieties within the template may explain the inability of FD31-Lys-Checker to nucleate calcite.

Our results demonstrate that computational protein design now allows the control of proteins over mineral formation that occurs during biomineralization to be studied with genetically encoded molecular templates that have known stable 3D structures, are chemically and structurally tunable, and can be engineered with atom-scale precision[39,52,53]. Compared to previous studies of natural and engineered proteins[11,12,31], using de novo designed proteins allows more rigorous testing of how structurally defined biomolecular surfaces control inorganic crystallization through systematic variations in the net charge, the patterning of charges, sidechain identity, solvent accessible surface area, and surface hydrophobicity. Current results suggest that designed proteins could enhance the biogenic formation of $CaCO_3$, an important carbon sink in the biogeochemical cycle[54]. More generally, our approach sets the stage for the programmable control of crystal polymorph, nucleation pathway, growth mechanism, and final crystal morphology, with the ultimate goal of developing hybrid materials with novel functions, including carbon sequestration.

## Methods
### Materials
*Escherichia coli* strains Lemo21(DE3) from New England Biolabs (NEB), BLR(DE3) from Novagen. All of Studier's autoinduction media components were bought from Sigma-Aldrich: tryptone, yeast extract,

potassium phosphate monobasic ($KH_2PO_4$), potassium phosphate dibasic ($K_2HPO_4$), potassium chloride (KCl), disodium hydrogen phosphate ($Na_2HPO_4$), potassium phosphate monobasic ($KH_2PO_4$), ammonium chloride ($NH_4Cl$), sodium sulfate ($Na_2SO_4$), iron (III) chloride ($FeCl_3$), calcium chloride ($CaCl_2$), manganese (II) chloride ($MnCl_2$), zinc sulfate monohydrate ($ZnSO_4·H_2O$), cobalt (II) chloride hexahydrate ($CoCl_2·6H_2O$), nickel (II) chloride ($NiCl_2$), copper (II) chloride ($CuCl_2$), sodium molybdate ($Na_2MoO_4$), sodium selenite ($Na_2SeO_3$), boric acid powder ($H_3BO_3$), α-D-glucose ($C_6H_{12}O_6$), D-lactose monohydrate ($C_{12}H_{22}O_{11}·H_2O$), magnesium sulfate ($MgSO_4$), and sodium chloride (NaCl).

The following components were also purchased from Sigma-Aldrich: kanamycin sulfate ($C_{18}H_{36}N_4O_{11}·H_2O_4S$), carbenicillin disodium ($C_{17}H_{16}N_2O_6S·2Na$), isopropyl β-D-1-thiogalactopyranoside (IPTG) ($C_9H_{18}O_5S$), tris(hydroxymethyl)aminomethane hydrochloride (Tris) ($NH_2C(CH_2OH)_3·HCl$), imidazole ($C_3H_4N_2$), 3-[(3-cholamidopropyl)dimethylammonio]-1-propanesulfonate hydrate (CHAPS) ($C_{32}H_{58}N_2O_7S·xH_2O$), phenylmethanesulfonyl fluoride (PMSF) ($C_7H_7FO_2S$), deoxyribonuclease I from bovine pancreas (DNAse), glycerol ($HOCH_2CH(OH)CH_2OH$), and polyethylene glycol 3350 ($H(OCH_2CH_2)_nOH$).

A Ni-NTA resin (Qiagen) was used for purification using immobilized metal affinity chromatography (IMAC). Tobacco Etch Virus (TEV) Protease was produced in-house, but is available elsewhere commercially (e.g., Sigma-Aldrich).

Calcium chloride dihydrate ($CaCl_2·2H_2O$, Sigma-Aldrich), sodium bicarbonate ($NaHCO_3$, Sigma-Aldrich), and ultrapure Milli-Q water ($H_2O$, 18.2 MΩ/cm) were used as the reagents for $CaCO_3$ synthesis experiments. Nuclease-free water was bought from Ambion. Tris buffer (pH 7) and Potassium Chloride (KCl) were bought from Sigma-Aldrich. Mica was bought from Ted Pella, CA.

## Designing FD31 protein
Protein design was carried out using the Rosetta Macromolecular Modeling Suite[55]. The protocol used is adapted from refs. 39 and 40, where initial structures were generated using RosettaRemodel followed by scoring with a coarse-grained function weighted with target helical parameters. We tested a number of combinations of sampled fragment lengths of helices and loops to then determine that the best was to sample helices of equal length or with a difference in length within three residues. Helix lengths of 16–30 residues were sampled. The helical parameters were biased to have a radius of 500 nm, a rise of 0 Å and a curvature of 0 rad, yielding a protein with a flat, repetitive surface. The inter-repeat distance was biased to 10.9 Å using harmonic constraints and allowing for a deviation of 0.05 Å. Sequence design was carried out using FastDesign with an enforced inner repeat symmetry and enforced interhelical repeat spacing preventing the backbone minimization cycles to create scaffolds with spacings deviating from the target. Residues on one side of the surface of the protein were mutated to glutamate manually or using PyRosetta[55].

## Designing DHR49-Neg protein
The molecule named DHR49 found in the published work of Brunette et al.[39] was used as a scaffold. It was turned into DHR49-neg by expanding it to 6 repeat units and mutating one of its helical surfaces to contain only negatively charged residues (24 aspartates and 18 glutamates). Lastly, the capping features were removed by mutating the sequence of the terminal repeats to the sequence of the internal repeats to produce head-to-tail interfaces containing the same hydrophobic interactions found in the core of the protein.

## Designing FD15 protein
This molecule is also reported in ref. 41. The design method is reported in the previous citation and is briefly described below: A helical

secondary structure element is placed along additional secondary structure elements that will be part of the repeating unit using BundleGridSampler mover in RosettaScripts. The rigid body transformation for the repeat propagation is set by translating a copy of the original helix; copies are added as needed. Degrees of freedom are limited to the helix phase, displacement of repeating helices on the XY plane, and change in height between adjacent helices. In the case of this molecule, the inter-repeat distance was set to 8.7 Å. Sequence design was carried out using FastDesign with an enforced inner repeat symmetry. Residues on one side of the surface of the protein were mutated manually or using a custom PyRosetta protocol to glutamates. The top protein designs were selected based on Rosetta metrics of protein energetics, core packing, secondary structure shape complementarity, helix quality, number of secondary structures in contact, and buried unsatisfied hydrogen bonds[55]. The sequences of these designs were submitted to previously reported structure prediction methods, and designs whose lowest energy predicted structure were within 2 Å root-mean-square deviation (RMSD) of the design model were selected for experimental characterization[40,56]. Scripts implementing the design protocols are available in the GitHub repository associated with this work (https://github.com/fatimadavila/DHR_CaCO3).

## Designed protein flattening protocol
We used a RosettaScripts protocol to create idealized (perfectly flat and repetitive) models of FD15, DHR49-Neg, and FD31 with a range of inter-repeat spacings to produce models suited for computationally screening for geometric matches between the proteins and calcite surfaces. First, the following constraints are applied: distance constraints to enforce distances between alpha-carbons in adjacent repeats (ranging from 8 Å to 15 Å with 0.1 Å increments), angle constraints to eliminate any curvature (by constraining the angle between repeated alpha-carbons in the first, middle, and last repeat is 180°), and dihedral constraints to prevent any twist (by constraining the torsion angle between two non-repeat c-alphas in the first repeat to and the corresponding two atoms in the last repeat to be 0°). Using a standard Rosetta score function (beta_nov16) with the addition of these constraints to score the models, we performed Monte Carlo sampling of backbone torsion angles while enforcing symmetry among the repeats to find the lowest energy model that satisfies these constraints, and thereby is perfectly flat and repetitive. We determined the repeat distance that produced the lowest energy model for each DHR (8.7 Å for FD15, 10.6 Å for DHR49-Neg, and 11.4 Å for FD31), and selected the models with repeat distances constrained to within 1 Å of these values for docking on calcite surfaces. RosettaScript XML files implementing the protocol and the resulting models are available on GitHub. (https://github.com/fatimadavila/DHR_CaCO3).

## Modeling protein on CaCO$_3$ surfaces
The idealized DHR models were then placed onto models of calcite {110}, {202}, and {104} and vaterite {010} surfaces and docked with Rosetta. Each docking trajectory starts by randomly rotating the protein on the surface. This starting pose is input into a Monte Carlo protocol that samples both the rigid body orientation of the protein relative to the surface and interfacial sidechain rotamers. The side chains are sampled in such a way that the same rotamer is placed at every equivalent repeat position. Finally, the binding score in Rosetta Energy Units (R.E.U.) is calculated by subtracting the median score of models of the protein and surface separated by 10 nm from the score of the protein docked onto the surface. Models of each protein on each surface were selected based on their total beta_nov16 Rosetta score to consider both the strain of the protein required to adopt the particular repeat spacing and the protein surface interactions. Implementations of this Rosetta protocol and the resulting models are available on GitHub. (https://github.com/fatimadavila/DHR_CaCO3).

## Expression and purification of proteins

Genes encoding the designs were then ordered through Genscript. Constructs with a N-terminal His6-tag followed by a TEV cleavage site were cloned into either pET-28b+ or pET21b between NdeI and XhoI sites. An additional flexible linker with a tryptophan was added to help with protein concentration determination by absorbance at 280 nm. The cloned genes were transformed into either Lemo21(DE3) *E. coli* from New England Biolabs (NEB) or in BLR(DE3) *E. coli* cells from Novagen. Expression then proceeded for 24 h at 37 °C using 0.5 L cultures in 2 L flasks using Studiers M2 autoinduction media with 50 μg/mL kanamycin or 50 μg/mL carbenicillin for pET-28b+ or pET21b, respectively. This expression protocol was selected because it gave higher yields than overnight autoinduction at 25 °C or induction with IPTG (at OD600 between 0.4 and 0.8) for either 4 h at 37 °C or 18 h at 18 °C.

Cells were pelleted at $4000 \times g$ for 30 min at 12 °C, then resuspended in ≈ 40 mL of lysis buffer (20 mM Tris, 500 mM NaCl, 30 mM imidazole, 0.25% (w/v) CHAPS, 1 mM PMSF, 1 mg/mL DNAse, pH 8) and finally lysed after homogenization using a microfluidizer (Microfluidics M110P) at 18 K pounds force per square inch. The lysate was clarified at $24,000 \times g$ for 30 min at 12 °C, and the soluble fraction was filtered through 0.7-μm syringe filters and set to do overnight batch binding at 4 °C with 1.5 mL of Ni-NTA resin (Qiagen) equilibrated in wash buffer (20 mM Tris, 500 mM NaCl, 30 mM imidazole, 0.25% (w/v) CHAPS, 5% (v/v) glycerol, pH 8). This was then transferred to a gravity column and washed with 25 mL of wash buffer before elution in 3 mL of elution buffer (20 mM Tris, 500 mM NaCl, 500 mM imidazole, 0.25% (w/v) CHAPS, 5% (v/v) glycerol, pH 8). Eluate was then dialyzed in 3.5 kDa molecular weight cut-off dialysis cassettes (Thermo) into 5 L of TEV cleavage buffer (50 mM Tris, 50 mM NaCl, pH 8.0) three times, each time for 2 h at room temperature (25 °C), before starting overnight cleavage at 4 °C by adding TEV protease in a ratio of 1 mg for each 25 mg of tagged protein. Secondary IMAC was carried out to remove the TEV protease and uncleaved product. The flowthrough was collected for fractionation by size exclusion chromatography with an AKTA pure chromatography system on a Superdex 200 Increase 10/300 GL column in TBS (20 mM Tris pH 8.0, 100 mM NaCl). The purified proteins were then dialyzed into MOPS buffer (10 mM) adjusted to pH 7. Dialysis was carried out three times overnight at 4 °C (i.e., changing the buffer every 24 h) with a dialysis ratio of 1:10,000 volume each time. Protein concentration was adjusted to 7 mg/mL, and 20 μL aliquots were snap-frozen for long-term storage. DHR49-Neg was purified using a different protocol described in ref. 57.

## Measuring protein concentration

Absorbance at 280 nm wavelength of 2 μL of protein samples was measured using a Nanodrop 8000 spectrometer (Thermo Scientific). The concentration was then calculated based on the measured absorbance and the known extinction coefficient following the Beer-Lambert law.

## Circular dichroism

Using a Jasco J-1500 CD spectrometer, measurements were taken on a sample with a concentration of 0.3 mg/mL in 20 mM Tris pH 8 and 100 mM NaCl, using a 1 mm path length cuvette. The raw CD signal was divided by $N \times C \times L \times 10$ to convert it to mean residue ellipticity, where $N$ is the number of residues, C is the protein concentration, and L is the path length of 0.1 cm.

## SAXS

SAXS data was acquired at the SIBYLS High Throughput facility, located at the Advanced Light Source in Berkeley, California[58]. Samples underwent 0.3-second exposures to the X-ray beam for a total of 10.2 seconds to generate 33 frames per sample. Both low (1 mg/mL)

and high (5 mg/mL) protein concentrations were used in a buffer containing 25 mM Tris pH 8.0, 150 mM NaCl, and 2% (v/v) glycerol.

Data analysis and determination of the best dataset was performed using the "SAXS FrameSlice" tool available on the SIBYLS website[59]. In cases where evidence of aggregation was seen in any of the 33 frames, only data preceding the aggregation were utilized in the Gunier region, and all data were included if there was no sign of aggregation. All collected datasets were analyzed for the Wide and Porod regions. The resulting experimental dataset was compared to a profile predicted from the design model using the FoxS SAXS server[60].

## X-ray protein crystallography

Crystallization trials employed sitting drop vapor diffusion with 200 nL drops in a 96-well plate format at a temperature of 20 °C. A Mosquito LCP from SPT Labtech was used to set up the plates which were then imaged with JAN Scientific UVEX and UVEX PS-256 microscopes. Diffraction-quality crystals were obtained in drops containing 0.1 M Tris pH 6.5 and 25% (w/v) polyethylene glycol 3350, rapidly frozen in liquid nitrogen, and sent to the synchrotron.

The Advanced Photon Source beamline on 24-ID-C was used to collect diffraction data. XDS was used to assess and integrate X-ray intensities and data reduction[61] and Pointless/Aimless in the CCP4 program suite was used to merge/scale them[62]. The process of structure determination and refinement began with molecular replacement in Phaser[63] using the protein design model. Next Phenix AutoBuild was used to improve the model[64], with rebuild-in-place set to false and using simulated annealing and prime-and-switch phasing to mitigate model bias. Phenix was used for refinement of the structure[64], and COOT for model building[65].

**CaCO₃ crystallization experiments.** In a typical crystallization experiment, 0.5 mL 10 mM $CaCl_2$ was mixed with 0.5 mL 10 mM $NaHCO_3$ as a control group. In the experimental group, we mixed 10 mM $CaCl_2$ with 2.16 μM protein first and then added 10 mM $NaHCO_3$ to initiate the nucleation and growth of $CaCO_3$. All protein stock solutions contained 7 mg/mL protein and were in 10 mM MOPS (pH 7).

## Ex situ TEM

Ex situ TEM samples were prepared by depositing 0.6 μL of reaction solutions, collected at various time points, onto a carbon-coated copper grid (300 mesh, obtained from Ted Pella) that was treated by plasma cleaning. The TEM analysis was conducted using a FEI Titan ETEM operating at 300 kV.

## Cryo-TEM

Cryo-TEM experiments were performed in a FEI Titan ETEM 80–300 kV to determine the particle morphology and size. Prior to the vitrification procedure, a pure lacey carbon grid was surface plasma-treated to make it hydrophilic. Using an automated vitrification robot (FEI Vitrobot Mark III, blot time: 3 s), a 3.0 μL sample extracted from a mineralization solution containing equal volumes of 10 mM $CaCl_2$, 2.16 μM protein, and 10 mM $NaHCO_3$ was loaded onto a grid and plunged into liquid ethane at about −183 °C. The vitrified sample was stored and transferred in a liquid nitrogen-cooled Gatan cryogenic holder operating near −196 °C.

## UV−Vis and DLS

The $CaCl_2$ incubated FD31 protein solution was measured by UV−Vis spectrophotometer (Ultrospec 2100 pro) and DLS (Malvern Zetasizer) to evaluate the protein-$Ca^{2+}$ interactions.

## In situ ATR-FTIR

A Bruker LUMOS II FTIR spectrometer equipped with Superior μ-ATR-FTIR capabilities was used to track $CaCO_3$ crystallization in real time. Precise control of the detector location within the reaction solution

was enabled by a retractable diamond crystal integrated in the lens which is manipulated by high-precision piezoelectric motors. Crystallization was initiated by combining 200 μL 10 mM NaHCO$_3$ and 200 μL 10 mM CaCl$_2$ in the presence or absence of proteins. Each FTIR spectrum was recorded with 8 scans at 2 cm$^{-1}$ resolution with H$_2$O as the background signal. The first spectrum was collected at ≈ 5 s after mixing the solutions.

### LP-TEM

Hummingbird Scientific liquid-cell chips contain two square silicon chips measuring 2.6 × 2.6 mm$^2$ each containing 50-nm-thick silicon nitride (Si$_3$N$_4$) membranes with 50 × 200 μm$^2$ imaging windows. Prior to use, the chips underwent a 2-min plasma cleaning process with a Harrick Plasma Cleaner. For a typical experiment, a solution containing 0.3 μL 10 mM CaCl$_2$ and 2.16 μM FD31 was applied to the spacer chip with a spacer thickness of 100 nm. Next, 0.3 μL of a 10 mM NaHCO$_3$ solution was added, and the reaction solution was sealed using a window chip to form a liquid-cell. The liquid-cell was then placed inside the Hummingbird Scientific sample holder. A leak check was conducted by inserting the assembled holder into a Hummingbird Scientific high-vacuum leak checking station for 2 min. This station features a low base pressure (<1 × 10$^{-6}$ mbar), short pumping and venting times, and allows the liquid-cell window to be imaged with optical microscopy. Immediately after the leak check the holder was inserted into a Thermo Fisher Scientific field emission Titan ETEM with an operating voltage of 300 kV. TEM images were captured after approximately 5 min using an Eagle CCD with a resolution of 1024 × 1024 pixels, a condenser aperture size of 50 μm, and spot size of 3. In situ movies were recorded using the free software Camstudio and post-processing of images from the movies was performed using the open-source software ImageJ. A low electron dose rate of ~100 e/nm$^2$s was employed to minimize beam effects.

### AFM

For DHR49-Neg, the protein stock solution was diluted to 0.5 μM in a 20 mM Tris buffer with 3 M KCl. Then 100 μL diluted protein solution was incubated on freshly cleaved mica for 30 min. For FD31, the protein stock solution was diluted to 1.0 μM in nuclease-free water with 5 mM CaCl$_2$. Then 100 μL diluted protein solution was incubated on freshly cleaved mica for 10 min. The as-assembled proteins on mica were imaged using Cypher-ES AFM (Asylum Research, CA) in a 20 mM Tris buffer with 3 M KCl, and 5 mM CaCl$_2$, respectively. The amplitude modulation mode and SNL-10-A probe (Bruker, CA) were used in the AFM experiments. The data processing was done with SPIP software (Image Metrology, Denmark).

### Reporting summary

Further information on research design is available in the Nature Portfolio Reporting Summary linked to this article.

## Data availability

Design models in protein data bank (PDB) format are available in a Github repository (https://github.com/fatimadavila/DHR_CaCO3/tree/main/dhr_models). The crystal structure of FD15 is available in the RCSB Protein Data Bank (pdb id: 8UGC). All data not included in the article and its Supplementary Information are available upon request from the corresponding authors.

## Code availability

The Rosetta Macromolecular Modelling suite is available for non-commercial use at https://www.rosettacommons.org. A GitHub repository (https://github.com/fatimadavila/DHR_CaCO3/) contains the Rosetta protocols used to design and model the proteins.

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

## Acknowledgements

Design, production, and biochemical characterization of initial proteins designs were performed at the Institute of Protein Design (IPD) in the Biochemistry Department at the University of Washington (UW) in Seattle WA with support from the Department of Energy (DOE) Office of Science (SC) Office of Basic Energy Sciences (BES) Biomolecular Materials Program under award DE-SC0018940. Subsequent design and production of modified designs was performed at the IPD in the Biochemistry Department at UW through support from the DOE SC BES, as part of the Energy Frontier Research Centers (EFRC) program: CSSAS, The Center for the Science of Synthesis Across Scales, under Award DE-SC0019288. Modeling of protein-CaCO₃ interfaces was supported by

the Audacious Project. Synthesis and characterization of CaCO₃, ATR-FTIR analysis, high-resolution and in situ LP-TEM, and AFM were performed at the Pacific Northwest National Laboratory (PNNL) with support from the DOE, SC, BESEFRC program through CSSAS under Award FWP 72448. PNNL is operated by Battelle for the Department of Energy under contract no. DE-AC05-76RLO1830. ATR-FTIR and TEM measurements were performed under user proposal 60575 at the Environmental Molecular Sciences Laboratory (EMSL), a DOE SC User Facility sponsored by the Office of Environmental and Biological Research under Contract No. DE-AC05-76RL01830. The AFM experiments were also conducted at the Molecular Analysis Facility (MAF), a National Nanotechnology Coordinated Infrastructure (NNCI) site at the UW, which is supported in part by funds from the National Science Foundation (awards NNCI-2025489, NNCI-1542101), the Molecular Engineering & Sciences Institute, and the Clean Energy Institute. Crystallographic data collected at the Northeastern Collaborative Access Team beamlines, which are funded by the National Institute of General Medical Sciences from the National Institutes of Health (P30 GM124165). SAXS analysis was performed at Advanced Photon Source; a U.S. DOE SC User Facility operated for the DOE SC by Argonne National Laboratory under contract no. DE-AC02-06CH11357. XRD data were obtained at the Advanced Light Source (ALS) of Lawrence Berkeley National Laboratory (LBNL), which is supported by the U.S. DOE BES under Contract No. DE-AC02-05CH11231.

## Author contributions

F.A.D.H., B.J., H.P., D.B., and J.J.D.Y. conceived the idea and designed the research; F.A.D.H., H.P., and T.H. designed, synthesized, and characterized the DHR proteins. B.J. designed and tested the CaCO₃ crystallization experiments using ex situ TEM, cryo-TEM, and in situ LP-TEM. B.J. and Z.W. ran the in situ ATR-FTIR measurements. S.Z. imaged the assembled complex via AFM. C.L.C. did the XRD test. A.K.B. and A.K. performed protein crystallization and X-ray structure determination. D.B. and J.J.D.Y. supervised the project. F.A.D.H., B.J., H.P., D.B., and J.J.D.Y. wrote the manuscript, and all authors contributed to revisions.

## Competing interests

The authors declare no competing interests.
