## [Peer Review File · Nature Communications]

Reviewers' Comments:

Reviewer #1:

Remarks to the Author:

This is a really well described study that has thoroughly explored the use of designed helical repeat proteins to nucleate and biotemplate calcium carbonate crystallization. The authors demonstrate the polymorph selectivity and crystallographic face selectivity of their designed proteins, and clearly show the range of conditions that these work in. They have used rational design to this, described the findings in detail. It is novel, significant, and thorough. There are small tweaks that would improve the experience of the reader and aid in reproducing the work and building upon it that are suggested below.

TLDR: check formatting on:

- Capitalization of materials etc.
- Abbreviations are explained upon first use.
- Writing on the figures is consistently too small to read.
- Superscripts and subscripts.
- Italicization of Latin phrases.
- Spacing between units and values is inconsistent.
- Random extra or missing spaces in text.
- Use correct symbols e.g. \approx means about, not \sim .
- Tenses of methods.
- Check the correct parentheses are used for the Miller notation.

Detailed comments:

Abstract:

"...the atomic structure of the organic-inorganic interfaces that regulate mineralization remains unknown⁶⁻⁸."

This is true for most biominerals (calcium carbonates, calcium phosphates, silica, magnetite, etc), but there has been some excellent X-ray work on human ferritins (<https://chemistry-europe.onlinelibrary.wiley.com/doi/10.1002/chem.202000064>) and the beginning of Cryo-EM work on bacterioferritins (<https://www.biorxiv.org/content/10.1101/2021.02.04.429857v1>) that show iron binding to the nucleation sites on the proteins, and the nascent mineralization process. Please alter to reflect this, either by specifying you are referring to calcium based biominerals, or that very little is known about the biomineral-protein interface. This is more clearly addressed in the phrasing in the introduction.

Introduction:

"For example, natural proteins nucleate specific calcium carbonate (CaCO₃) polymorphs^{13,14,20-22}, but none have known stable tertiary structures and many are intrinsically disordered or insoluble⁶⁻⁸."

This raises a question – why does nature choose these intrinsically disordered proteins or insoluble proteins, and you have chosen soluble, structured proteins? I assume it's because we can make and handle soluble structured proteins, but are not as good as a biomineralization organism at designing, making and using insoluble disordered biomineralization proteins. This should be clarified, nature doesn't use epitaxy in biomineral control, but we can do that. It leads into why you chose your design strategy.

"...self-assembled monolayers (SAM)³²" should be (SAMs)

Results:

This is a general comment. It is usually difficult to produce highly negatively charged proteins in *E. coli*, it is often incredibly low yield and the organisms struggle to grow well. Can some detail of the

strains and conditions chosen and used for this be added to the supplementary information, and how much optimization was required to find good conditions - were any strains / conditions tried that were rubbish? Some of the designs did not express & purify - were there any features (apart from the negative charge) that may have contributed to this? Could the data collected on the success / failure of designs from Rosetta be used to develop an empirical or machine learned intelligent "expressability index." This would be extremely valuable for improving success for future design work, and would researchers make good decisions in the future when selecting their molecular biology strategy, especially as many biomineralization proteins are negatively charged.

The writing on Figure 1a is too small to read easily. The designs in 1d, g and f are also quite small and difficult to see. Could the design models be shared as .pdb files in the supplement as they are not yet available as crystal structures on the protein data bank? Or links to the crystal structures of the designs be added. You describe protein crystallization and structure refinement in the methods, but do not make it clear if you are showing models, what the resolutions of the structures are, etc.

How did you select 40 designs for experimental characterization and two for detailed characterization? Randomly or with an experienced eye? Why did you select 40 / 2 respectively?

Add to the supplementary table S1 info showing the * expression levels (mg/mL culture) - what you count as high someone else might think is low or vice versa.

Were control experiments done where the same volume of MOPS buffer used to store the proteins in was added to the mineralization solution - did this have any effect? In iron oxide mineralization by co-precipitation, adding small amounts of buffer alters the products of the mineralization in the absence of protein. Calcium carbonate is nowhere near as sensitive, but this should be checked. The BSA control is sufficient if it is exactly the same as the conditions not adding any protein at all, but add in the buffer controls if you have them.

Were CD experiments done to check the protein stability as pH changes? If the Tris NaCl pH 8 is close to the expected pH of the mineralization conditions, this second question has been answered for the final mineralization conditions, but not for the initial incubation in CaCl₂ when the protein is thought to bind the calcium ions. It would be interesting to see if the protein becomes more or less structured at lower pH and in the presence of calcium.

Figure 2 caption: CaCO₃ needs appropriate subscripts adding back in, the formatting has removed this for some reason.

Latin should be italicized, e.g. *E. coli*, *ex situ*, *in vivo*.

Pg 4: 1 μ M should be 1 μ M, also was the BSA control in the same MOPS buffer as the proteins being tested?

Was any cryo-TEM or negative stain TEM done of the calcium incubated assemblies? This would nicely show some information on the protein structure as the mineral begins to form as shown in F3d. This is not necessary for this work, as it would be a significant undertaking, but could form some excellent detailed information on the protein-mineral interface for future work.

Figure S4 has an extra space after the "e" in "Panel e."

Pg6 "(Highlighted in Table S2)" should have a lowercase "h".

Figure S8 - This is oversimplified. In solution the amino acid side chains can adopt a number of different rotamers, with the longer side chains (such as glutamic acid in this case) being more likely to adopt a wider range of conformations. In a lot of proteins, these side chains will flip and alternate between a number of rotamer positions, which can get locked into an apparent preferred positions upon protein crystal formation (required for X-ray structure determination). This is an artefact that is less likely to manifest from Cryo-EM single particle structures, but is intrinsic to the majority of the protein structures on the PDB, as most of these were determined by X-ray protein

crystallography. It may be that, in the presence of mineral precursors, or when in contact with a mineral surface, there is a higher preference for adopting a preferred rotamer, as the presence of the mineral stabilizes a preferred side chain orientation, and this preferred orientation is different for glutamic acid vs aspartic acid due to the side chain length – I think this is more what you are meaning to say?

The labels on the supplementary TEM images (e.g. S7 & S9) are too small to read easily, and the writing seems to be much lower resolution than the labels shown in the main figures. The images could easily be made much larger so it is easier to see the features and labels you are showing, and should be exported at a better quality level so the writing can be read.

Pg 7. You have used the \sim symbol when I think you mean \approx . How were these measurements of particle size taken, how many crystals were counted – is it just the ones you show TEM images of, or is it from multiple images across multiple experiments, what's the error on these measurements?

Fig. S11 CaCl_2 needs a subscript 2.

Fig. S12 CaCO_3 all need subscript 3.

All the diagrams show the interface between calcite and the proteins that select calcite. You should look for matches (or lack thereof) to the vaterite faces. Especially for the checker examples, as one of these still selects calcite and the other selects vaterite, whereas the all glutamate versions both selected either the $\{110\}$ or $\{202\}$ calcite faces. If these have been checked and there is no match, add an example or two to help illustrate how good the match is to the calcite surfaces discussed.

Discussion:

Pg 9 double space at the start of the 3rd paragraph, "...this templating effect. We found that..."

Triple space at the start of the list in the 3rd paragraph, "...a pronounced effect (Table S2): FD15 with an array spacing of..."

double space before "...5, 8 and 11 nm..."

"...5, 8 and 11 nm..." Are these nm² or nm long? Please clarify.

there is a double space between a and linear "...crystals (there is a linear scaling..."

References – check the book refs are in full, some seem a little short of info e.g. on publisher and place published.

The proteins work within a "Goldilocks" range of calcium, carbonate and / or protein concentrations – can the authors comment on what this might mean for how natural biomineralization proteins might work in vivo in combination with other controlling factors. For example to help control polymorph / orientation / faces selected – does the presence of DHRs mean that the calcium concentration can be higher / lower, do you need a lot of the protein, does their co-localization (self-assembly) into larger structures offer better / worse control?

Supplementary information:

Where the supplementary Figures were referred to in the main text there are comments above, but additional comments are listed below. For some reason the referencing style is different in the supplementary information (X) numbers rather than superscript numbers. Also it would be good to add page numbers to the supplementary to make it easier to navigate.

Materials and Methods.

You used many more different chemicals and products than are listed in the materials section. Please complete this.

At the end of the designed protein flattening protocol section, there is an extra space between protocol and and, "...RosettaScript XML files implementing the protocol and the resulting models..."

Sometimes you have a space between the numbers and the units, sometimes you don't. For example "...1mM PMSF, 1 mg/mL DNase, pH 8..." should have a space between 1 and mM. This varies throughout and should be corrected in all cases. Please check your units are consistent with "per" for example, for using mg/mL or mg mL⁻¹.

The TEV cleavage buffer recipe is in size 12 rather than size 11 font, check your font size and type are consistent.

Specify if the % proportions in the buffer recipes are v/v w/v or w/w.

24,000 xg – needs the x in it and the space

dialyzed should be spelled in US English to be consistent with the rest of the manuscript.

How long was each of the 5 L dialysis against TBS carried out for, and at what temperature – room temperature or 4 °C? Also the same for the final dialysis into 10 mM MOPS buffer – you say 3 times overnight, so is that 3x 24 hours, or were you in the lab changing over the buffer solutions every 4 hours through the night? Or are you lucky enough to have a machine to do this for you?

Check your capitalizations are correct: cleavage, pET21b, imidazole, CHAPS,

Explain your abbreviations the first time you use them, e.g. IMAC, CHAPS, MOPS, etc. You have done this with some and not with others.

Cryo-TEM – "...lacey carbon grid is surface plasma..." should be ...was surface plasma... double check you have got all your tenses right throughout the work.

Did you do the cryo-TEM with different concentrations of proteins – what were these concentrations? Also, elsewhere you specify different concentrations of calcium and bicarbonate that you say were used in Cryo-TEM samples – why? Or correct it if it's not correct. The temperature of the liquid ethane should be below what temperature? I don't know what you mean by "...throughout the holder."

LP-TEM. Change the comma to a full stop and add a .These, "...0 × 200 μm² windows for imaging, were plasma cleaned..." How were they leak checked? How long did the leak check take?

Supplementary text

Number density estimation:

Check all your CaCO₃ etc. have the subscript. How you have written the very small numbers is not very clear. It would be better to express 1.85E17 nm³ as 1.85 × 10¹⁷ nm³.

Proposed mechanisms of protein control over calcite nucleation:

2nd paragraph begins with a space.

Spaces are wrong – 2 between matches and (202) and no space between (202) and (Fig. S9). "...better matches (202)(Fig. S9)."

Fig S2. (d) LP-TEM image shows ≈ 5 nm CaCO₃

Wherever you discuss the faces, you use the circular parentheses which indicates a specific face. For example (202) is a face, {202} are (202)-like faces (e.g. (022), and (202) are in the same

family in calcite, as are some of their negative intergers). Double check you have the notation correct for the face vs the faces in the main text and the supplementary. I noticed this in Fig. S7, but it happens consistently throughout the manuscript. [xxx] for a specific zone axis, and <xxx> for a family of directions, but I don't think this one crops up. You've done it right on the annotations on the TEM images (I think, as mentioned above the resolution is not good enough to read the tiny writing well, but they should probably be the {xxx} ones on the images, and possibly the specific (xxx) ones on the SAED/FFT's if the angles and everything was checked and matched properly).

Reviewer #2:

Remarks to the Author:

Designed proteins (DHRs) with flat surfaces displaying functional groups that lattice matched to calcite was used to modulate mineralization. The structural regularity and tunability promote polymorphic specific CaCO₃ nucleation. The authors design a variety of proteins to demonstrate the specificity of their system to drive biomineralization with rational genetic modular design. The manuscript demonstrates that spacing of the carboxyl sidechains can be used to drive different crystal formation. This is a well written manuscript and should be accepted for publication. The results are noteworthy and provides an approach to control surface chemistry to drive biomineralization.

Reviewer #1 (Remarks to the Author):

This is a really well described study that has thoroughly explored the used of designed helical repeat proteins to nucleate and biotemplate calcium carbonate crystallization. The authors demonstrate the polymorph selectivity and crystallographic face selectivity of their designed proteins, and clearly show the range of conditions that these work in. They have used rational design to this, described the findings in detail. It is novel, significant, and thorough. There are small tweaks that would improve the experience of the reader and aid in reproducing the work and building upon it that tare suggested below.

Response: Thank you for your feedback and thoughtful comments. They are addressed point by point below alongside descriptions of corresponding modifications to the manuscript.

Our responses and quotations from the text are in blue, with additions highlighted yellow.

TLDR: check formatting on:

- Capitalization of materials etc.

Corrected

- Abbreviations are explained upon first use.

Corrected

- Writing on the figures is consistently too small to read.

Response: We have adjusted all of the figures for legibility. The text size was set to the maximum font size allowable by this journal's figure requirements (arial size 7) whenever possible, and to size 6 in a few cases. This is within the 5-7 point font standards set by the journal. <https://www.nature.com/documents/NRJs-guide-to-preparing-final-artwork.pdf>

- Superscripts and subscripts.

Corrected

- Italicization of Latin phrases.

Corrected

- Spacing between units and values is inconsistent.

Corrected

- Random extra or missing spaces in text.

Corrected

- Use correct symbols e.g. \approx means about, not \sim .

Corrected \sim to \approx in main text and SI

- Tenses of methods.

Corrected

- Check the correct parentheses are used for the Miller notation.

Response: Thank you for carefully reading the manuscript and providing us with this feedback. We have addressed them in the updated version, see the details below.

Detailed comments:

Abstract:

“...the atomic structure of the organic-inorganic interfaces that regulate mineralization remains unknown^{6–8}.”

This is true for most biominerals (calcium carbonates, calcium phosphates, silica, magnetite, etc), but there has been some excellent X-ray work on human ferritins (<https://chemistry-europe.onlinelibrary.wiley.com/doi/10.1002/chem.202000064>) and the beginning of Cryo-EM work on bacterioferritins (<https://www.biorxiv.org/content/10.1101/2021.02.04.429857v1>) that show iron binding to the nucleation sites on the proteins, and the nascent mineralization process. Please alter to reflect this, either by specifying you are referring to calcium based biominerals, or that very little is known about the biomineral-protein interface. This is more clearly addressed in the phrasing in the introduction.

Response: Thanks for pointing out these studies, they are very interesting and very relevant! We have made the following highlighted edit in the abstract:

“but the atomic structure of the organic-inorganic interfaces that regulate mineralization remain **largely** unknown”,

And we also added a citation to the ferritin example you provided to the introduction:

In further support of this concept, the structures of ice-binding proteins contain surfaces with an epitaxial-like lattice matching the ice lattice³⁵, which enables modulation of ice formation by binding ice nuclei through preorganized ice-like waters, and ferritin has been shown to organize iron clusters on its interior surface³⁶.

Introduction:

“For example, natural proteins nucleate specific calcium carbonate (CaCO₃) polymorphs^{13,14,20–22}, but none have known stable tertiary structures and many are intrinsically disordered or insoluble^{6–8}.”

This raises a question – why does nature chose these intrinsically disordered proteins or insoluble proteins, and you have chosen soluble, structured proteins? I assume it's because we can make and handle soluble structured proteins, but are not as good as a biomineralization organism at designing, making and using insoluble disordered biomineralization proteins. This should be clarified, nature doesn't use epitaxy in biomineral control, but we can do that. It leads into why you chose your design strategy.

Response: We indeed work with soluble structured proteins because we can produce and handle them better than native proteins, and tune their shape and surface chemistry for experiments. This is conveyed in the manuscript in the text you quote above, and again in the discussion as follows:

“Compared to previous studies of natural and engineered proteins^{14,15,31}, using de novo designed proteins allows more rigorous testing of how structurally defined biomolecular surfaces control inorganic crystallization through systematic

variations in the net charge, pattern of charges, side chain identity, solvent accessible surface area, and surface hydrophobicity.”

As to why nature uses different kinds of proteins, evolution is a stochastic process with no incentive for using mechanisms with decipherable structures. As native protein-mineral interfaces are not structurally characterized (except for ferritin as you noted), and because native biomineralization proteins often operate in complex environments (e.g. on the surface of β -chitin or membrane bound organelles), some of which form beta-sheet aggregates with periodic structures (albeit ones that are not tractable for structural characterization), we don't feel comfortable making strong statements about what role epitaxy does or does not play in native biominerals. We are making the point that because we don't know what native biomineral interfaces look like, we tested a strategy informed by ice-binding proteins and SAMs.

To further address this topic, we have added the following section to the supplementary discussion:

Relevance of structured templates to native systems

While numerous native proteins associated with biomineral formation are inherently disordered proteins (IDPs)⁷⁰, there are a number of examples of both structured proteins and proteins that are self-assembled into ordered structures^{71,72} or that must interact with a preorganized scaffold⁷³ before mineralization occurs. Moreover, even in the case of IDPs, whether or not they become structured and/or organized by the time they form the protein-mineral interface is largely unknown.

“...self-assembled monolayers (SAM)³²” should be (SAMs)

Response: Changed accordingly. Please see it on the Page 2: “Stereochemical-matched self-assembled monolayers (SAMs)³²”

Results:

This is a general comment. It is usually difficult to produce highly negatively charged proteins in E. coli, it is often incredibly low yield and the organisms struggle to grow well.

Response: Thanks for raising this interesting issue.

Our success in purifying these proteins is likely related to their idealized structures. In our experience, highly negatively charged de novo proteins have been easier to produce than ones with positive or neutral charges. We speculate that the discrepancy between our experience and the reviewer's comments may be that they are referring to unstructured Asp-rich proteins that have been identified in native biominerals.

Can some detail of the strains and conditions chosen and used for this be added to the supplementary information, and how much optimization was required to find good conditions - were any strains / conditions tried that were rubbish?

Response: We added the highlighted description of alternate growth conditions we tried:
“Expression then proceeded for 24 hours at 37 °C using 0.5 L cultures in 2L flasks using Studiers M2 autoinduction media with 50 µg/mL kanamycin or 50 µg/mL carbenicillin for pET-28b+ or pET21b, respectively. **This expression protocol was selected because it gave higher yields than overnight autoinduction at 25 °C or induction with IPTG (at OD600 between 0.4 and 0.8) for either 4 hours at 37 °C or 18 hours at 18 °C.**”

Some of the designs did not express & purify – were there any features (apart from the negative charge) that may have contributed to this? Could the data collected on the success / failure of designs from Rosetta be used to develop an empirical or machine learned intelligent “expressability index.” This would be extremely valuable for improving success for future design work, and would researchers make good decisions in the future when selecting their molecular biology strategy, especially as many biomineralization proteins are negatively charged.

Response: Learning the sequence and structural requirements for soluble expression from experimental data is a very interesting avenue of research that is outside the scope of this work.

The writing on Figure 1a is too small to read easily. The designs in 1d, g and f are also quite small and difficult to see. Could the design models be shared as .pdb files in the supplement as they are not yet available as crystal structures on the protein data bank? Or links to the crystal structures of the designs be added. You describe protein crystallization and structure refinement in the methods, but do not make it clear if you are showing models, what the resolutions of the structures are, etc.

Response: Changes to Figure 1 were carried out accordingly. Text was set to the maximum allowable size by the journal's figure requirements (arial size 7) whenever possible, and to size 6 when not. This is within the 5-7 pt guidelines from the journal.

The figure 1 caption explicitly states which images are of models vs. structures:

“**Design models** for (d) FD15 and (g) FD31. (e, h) Their respective circular dichroism scans showing mean residue ellipticity (M.R.E.) from 200 to 260 nm at 25 °C and 95 °C. (f) The **crystal structure of FD15 in gray** overlaid on the **design model in green** (solved at 3 Å resolution, RMSD of 0.45 Å).”

All of the protein models are available in the github repo associated with the publication at https://github.com/fatimadavila/DHR_CaCO3. The FD15 crystal structure is being deposited in the protein database (pdb id: 8UGC).

How did you select 40 designs for experimental characterization and two for detailed characterization? Randomly or with an experienced eye? Why did you select 40 / 2 respectively?

Response: The 40 designs selected for experimental characterization were selected as described in the methods:

“The top protein designs were selected based on Rosetta metrics of protein energetics, core packing, secondary structure shape complementarity, helix quality, number of secondary structures in contact, and buried unsatisfied hydrogen bonds (1). The sequences of these designs were submitted to previously reported structure prediction methods and designs whose lowest energy predicted structure were within 2Å root-mean-square deviation (RMSD) of the design model were selected for experimental characterization(3, 5).”

We have modified the main text to include the following description:

“We selected 40 designs for experimental characterization **following criteria described in the methods section** and obtained synthetic genes encoding them (Sup Table 1).”

The two that were selected for detailed characterization displayed the best monodispersity as assayed by SEC. We have incorporated these small additions in the main text:

“Eight of these expressed at **sufficiently** high levels in *E. coli* and were highly soluble. Two designs, FD15 and FD31, were selected for detailed characterization **because they displayed the best monodisperse behavior in solution as evidenced by size exclusion chromatography (SEC).**”

*Add to the supplementary table S1 info showing the * expression levels (mg/mL culture) – what you count as high someone else might think is low or vice versa.*

Response: Thanks for pointing this out. Our average yield is ≈ 20 mg per L of culture (after purification). We have included this information in the Table S1 caption.

Were control experiments done where the same volume of MOPS buffer used to store the proteins in was added to the mineralization solution – did this have any effect? In iron oxide mineralization by co-precipitation, adding small amounts of buffer alters the products of the mineralization in the absence of protein. Calcium carbonate is nowhere near as sensitive, but this should be checked. The BSA control is sufficient if it is exactly the same as the conditions not adding any protein at all, but add in the buffer controls if you have them.

Response: The no protein control did not contain MOPS, but the BSA control and the designed protein stock solutions were both stored at 7 mg/mL in 10 mM MOPS pH 7. In the crystallization reactions the protein stock solutions were diluted approximately 100 fold, so the final concentration of MOPS is around 0.1 mM. Therefore we do not expect the trace amount of buffer to be driving the observed effects on the nucleation of calcium carbonate.

We made the following correction and addition the methods section:

In a typical crystallization experiment, 0.5 ml 10 mM CaCl₂ was mixed with **0.5 ml 10 mM NaHCO₃** as a control group. In the experimental group, we mixed 10 mM CaCl₂ with 2.16 μM protein first and then added 10 mM NaHCO₃ to initiate the nucleation and growth of CaCO₃. **All protein stock solutions contained 7 mg/mL protein and were in 10 mM MOPS pH 7.**

Were CD experiments done to check the protein stability as pH changes? If the Tris NaCl pH 8 is close to the expected pH of the mineralization conditions, this second question has been answered for the final mineralization conditions, but not for the initial incubation in CaCl₂ when the protein is thought to bind the calcium ions. It would be interesting to see if the protein becomes more or less structured at lower pH and in the presence of calcium.

Response. We only ran CD in the pH 8 20 mM Tris buffer with 100 mM NaCl. The mineralization concentrations (5 mM CaCl₂ 5 mM NaHCO₃) and the pre-incubation conditions (10 mM CaCl₂) are both close to neutral pH. The small difference between these pH's and the pH 8 condition we ran in CD is unlikely to affect the charge and/or stability of the protein.

We expect that any small difference in the stability based on pH would be overshadowed by the effects of the Ca²⁺ ions themselves, especially as they cause the formation of Ca²⁺ assemblies. Running a CD on FD31 with Ca²⁺ would be interesting, but better suited for work that more fully interrogates the structure of the assemblies with cryo-TEM, as discussed below.

Figure 2 caption: CaCO₃ needs appropriate subscripts adding back in, the formatting has removed this for some reason.

Response: Corrected subscripts in the caption.

Latin should be italicized, e.g. E. coli, ex situ, in vivo.

Response: We revised this in the re-submitted version.

Pg 4: 1 uM should be 1 μM, also was the BSA control in the same MOPS buffer as the proteins being tested?

Response: Corrected uM to μM. Yes, the BSA control was in the same MOPS buffer as the proteins being tested.

We made the following correction and addition the methods section:

"In a typical crystallization experiment, 0.5 ml 10 mM CaCl₂ was mixed with 0.5 ml 10 mM NaHCO₃ as a control group. In the experimental group, we mixed 10 mM CaCl₂ with 2.16 μM protein first and then added 10 mM NaHCO₃ to initiate the nucleation and growth of CaCO₃. All protein stock solutions contained 7 mg/mL protein and were in 10 mM MOPS pH 7."

Was any cryo-TEM or negative stain TEM done of the calcium incubated assemblies? This would nicely show some information on the protein structure as the mineral begins to form as shown in F3d. This is not necessary for this work, as it would be a significant undertaking, but could form some excellent detailed information on the protein-mineral interface for future work.

Response: We are currently attempting this in order to understand the protein-mineral interface structure, but consider this beyond the scope of this project. The experimental result that so far yielded the most information is the unstained TEM micrograph shown in Fig. S5i. We did not perform cryo-TEM or negative stained TEM because regular ex situ TEM and LP-TEM provided enough contrast due to the involvement of Ca^{2+} in the assemblies.

Figure S4 has an extra space after the “e” in “Panel e.”

Response: Corrected. (this is now Fig. S5 because of reordering to reflect their order mentioned in the text.

Pg6 “(Highlighted in Table S2)” should have a lowercase “h”.

Response: Corrected.

Figure S8 – This is oversimplified. In solution the amino acid side chains can adopt a number of different rotamers, with the longer side chains (such as glutamic acid in this case) being more likely to adopt a wider range of conformations. In a lot of proteins, these side chains will flip and alternate between a number of rotamer positions, which can get locked into an apparent preferred positions upon protein crystal formation (required for X-ray structure determination). This is an artefact that is less likely to manifest from Cryo-EM single particle structures, but is intrinsic to the majority of the protein structures on the PDB, as most of these were determined by X-ray protein crystallography. It may be that, in the presence of mineral precursors, or when in contact with a mineral surface, there is a higher preference for adopting a preferred rotamer, as the presence of the mineral stabilizes a preferred side chain orientation, and this preferred orientation is different for glutamic acid vs aspartic acid due to the side chain length – I think this is more what you are meaning to say?

Response: Thanks for pointing out this issue. We have deleted this figure (formerly Fig. S8) as it does not contribute significantly to the discussion and is a source of confusion.

To your point, yes, the different side chains could be locked in a different preferred rotamer upon interaction with the mineral or ions in solution. The locked rotamers could then bias the formation of a facet by matching the carbonate orientation on it. Although it is possible/probable that the rotamers are locked into on conformation, we don't currently have evidence to conclude this. Therefore, we now state more generally that Asp and Glu have different sidechains and different rotamer sets, and this effects the possible interactions between the protein and the surfaces.

We deleted Fig. S8 and modified the corresponding text in the supplemental discussion to the following:

“Firstly, the use of odd (Asp) vs. even (Glu) number of carbons in the side chains of the binding moieties could bias the formation of different interfaces, as seen with self-assembled monolayers(16). Although both Asp and Glu sidechains adopt multiple rotamers, the possible orientations of their carboxyl groups relative to their backbone

atoms (and therefore the interactions they can make with a given surface) are distinct. This may contribute to the stabilization of different interfaces by FD31, which exclusively presents Glu and stabilizes calcite {110}, and DHR49-Neg which includes Asp residues and stabilizes calcite {202}. This stereochemical explanation is supported by the observation that mutating Glu residues to Asp ablates the calcite nucleation by FD31, whereas mutating Glu residues to isosteric Gln does not (Fig. 5d-g)."

We think that, especially considering the results of the Glu->Asp mutations shown in Fig. 5, some discussion of side-chain length is required, but we agree that our previous explanation was too simplistic and have made these changes to rectify that.

The labels on the supplementary TEM images (e.g. S7 & S9) are too small to read easily, and the writing seems to be much lower resolution than the labels shown in the main figures. The images could easily be made much larger so it is easier to see the features and labels you are showing, and should be exported at a better quality level so the writing can be read.

Response: Thanks for pointing this out. We have improved the image and figure quality (The formerly named Fig. S9 is now Fig. S8).

Pg 7. You have used the ~ symbol when I think you mean ≈. How were these measurements of particle size taken, how many crystals were counted – is it just the ones you show TEM images of, or is it from multiple images across multiple experiments, what's the error on these measurements?

Response: Thanks for pointing out this problem. We changed ~ to ≈. We are not trying to follow the size evolution, and thus did not analyze the size distribution over time in Figure 4. The size description is based on the representative TEM images in Figure 4. Based on the TEM image in the Fig. 4b, we measured the size (12 particles) to be $\approx 22.8 \pm 3.9$ nm and updated it in the new version of the figure caption:

"(b) *Ex situ* TEM image showing calcite grown through particle attachment. The mean measured particle size of the 12 particles shown is 22.8 ± 3.9 nm."

We also provided the particle number and particle size in the new version of the figure 5 caption:

"(a-c) TEM and the size distribution of particles formed in the presence of (a) FD31-Rep3 with 3 repeats, (b) FD31 with 6 repeats, and (c) FD31-Rep9 with 9 repeats. The corresponding particle number (N) and mean measured particle size (S) are: (a) $N = 172$, $S = 9.1 \pm 2.9$ nm, (b) $N = 150$, $S = 5.5 \pm 0.9$ nm, (c) $N = 147$, $S = 4.7 \pm 0.8$ nm, respectively."

Fig. S11 CaCl₂ needs a subscript 2.

Response: We corrected this in the new version (now Fig. S10)

Fig. S12 CaCO₃ all need subscript 3.

Response: We corrected this in the new version (now Fig. S11)

All the diagrams show the interface between calcite and the proteins that select calcite. You should look for matches (or lack thereof) to the vaterite faces. Especially for the checker examples, as one of these still selects calcite and the other selects vaterite, whereas the all glutamate versions both selected either the {110} or {202} calcite faces. If these have been checked and there is no match, add an example or two to help illustrate how good the match is to the calcite surfaces discussed.

Response: These are good points! Based on this feedback, we added the mutated FD31 variants (FD31-Asp, -Lys, and -Gln) proteins and the vaterite {010} face to the set. We revised Fig. S13 and Fig. S14 to show results from the all-by-all comparison of the total set of 6 proteins and 4 faces. The 3D coordinates of these models are included in the github repo, along with >600 other conformations of each protein on each surface for interested readers.

Deciding how to present these models of the interfaces has been challenging. We set out to look for potential lattice matching arrangements, as has been done with SAMs and other materials. Rosetta lets us efficiently sample arrangements of the proteins on the facets and screen them for geometric matching, using a score function that considers Van der Waals and electrostatic interactions between point charges. When we do this, and in particular when we vary the repeat-repeat spacings (as justified in the supplementary discussion and Fig. S3), we find a lot of lattice matches! Both between proteins and faces that experiments suggest form energetically favorable interfaces and those that do not.

For the cases where we have evidence that a particular protein stabilizes a particular facet, a docked conformation that geometrically matches complementary charges between the protein and the mineral were found, and represent our current best idea of what these interfaces may look like.

The fact that we also see lattice matches between other proteins and other surfaces (or even most proteins and most surfaces), tells us that the behavior of proteins at mineral-water interfaces is complicated, and beyond our current ability to model predictively. Unlike interfaces between biomolecules, there are no large training sets for protein-mineral interactions, at least not yet!

We have attempted to accurately describe this as follows in the supplementary discussion:

“Secondly, geometric lattice matching may play a role. DHRs with different repeat spacings (Fig. S3a-c) affect CaCO_3 growth differently (Fig. 2), and suggest interactions between specific proteins and specific facets (Fig. 4. Fig. S7). Rosetta docking simulations were run with DHR models constrained to be flat and repetitive to identify potential geometric matches at the protein-mineral interfaces (see methods). Since the constrained DHR models with repeat spacings surrounding the minima had comparable predicted energies (Fig. S3d), models of FD15, DHR49-neg, FD31, and FD31 variants (Fig. 5) with repeat distances constrained to $\pm 1 \text{ \AA}$ from the minima at 0.1 \AA intervals (to represent the possibility of small conformational changes in the proteins) were docked onto models of

three calcite facets and one vaterite surface while maintaining repeat symmetry in the protein. The {110} and {202} calcite facets were included based on the observed orientations of nanoparticles formed in the presence of FD31 and DHR49-Neg, respectively, as were the typically expressed calcite {104} and vaterite {010} surfaces. This Rosetta modeling (Fig. S13, S14) did not accurately capture the complex energetics of protein-mineral interactions at solid-liquid interfaces, discriminate between DHRs that nucleate or do not nucleate calcite (Fig. 2), explain the stabilization of specific facets (Fig. 4c, d, Fig S7), or predict the effects of the FD31 surface mutations (Fig. 5d-i). It was consistent with observations that the calcite nucleating DHR proteins bind calcite {110} and {202} rather than the {104} facet (Fig. 4c, d, Fig. S7, S13). Lattice-matching docks were observed both for interfaces that our results suggest template nucleation (e.g. DHR49-Neg on {202} and FD31 on {110}), and those that do not (e.g. FD15 on calcite {110} and FD31 on vaterite {010}; Fig. S14). However, these models do demonstrate that DHR proteins are structurally well suited to lattice match CaCO₃ surfaces, and to our knowledge the models of DHR49-Neg on calcite {202} and FD31 on calcite {110} are among the best supported models of protein-CaCO₃ interfaces that drive heterogeneous nucleation currently available. These models are provided in a GitHub repository alongside the scripts that produced them (https://github.com/fatimadavila/DHR_CaCO3).

Discussion:

Pg 9 double space at the start of the 3rd paragraph, "...this templating effect. We found that..."

Response: We corrected this in the new version.

Triple space at the start of the list in the 3rd paragraph, "...a pronounced effect (Table S2): FD15 with an array spacing of..."

Response: We corrected this in the new version.

double space before "...5, 8 and 11 nm..."

Response: We corrected this in the new version.

"...5, 8 and 11 nm..." Are these nm² or nm long? Please clarify.

Response: These are nm long. We've modified the text to reflect this.

there is a double space between a and linear "...crystals (there is a linear scaling..."

Response: We corrected this in the new version.

References – check the book refs are in full, some seem a little short of info e.g. on publisher and place published.

Response: References should be correct now, thanks for pointing this out. Kindly note that Nature style references do not require to state the place published.

The proteins work within a “Goldilocks” range of calcium, carbonate and / or protein concentrations – can the authors comment on what this might mean for how natural biomineralization proteins might work in vivo in combination with other controlling factors. For example to help control polymorph / orientation / faces selected – does the presence of DHRs mean that the calcium concentration can be higher / lower, do you need a lot of the protein, does their co-localization (self-assembly) into larger structures offer better / worse control?

Response:

While we would expect there to be “a “Goldilocks” range of the reagent concentrations, we did not explicitly explore any of these parameters other than protein concentration. However, we can certainly speculate on what the effect would be of varying the solute or protein concentration. This speculation is based on the physical picture that underlies our approach; namely, that a protein template can promote nucleation of a specific phase by lowering the interfacial free energy and thereby the free energy barrier to nucleation (Fig. 1a). The full explanation is a long one and certainly is not appropriate for the manuscript, but it provides a mechanistic understanding for why a Goldilocks range should exist.

The free energy barrier to nucleation depends on the ratio of the interfacial energy (γ) cubed divided by the supersaturation (s) squared. The supersaturation is defined as the product of the Ca^{2+} and CO_3^{2-} activities divided by the equilibrium constant (K_{sp}) and the activities of the two species are proportional to their absolute concentrations, where the proportionality constant is the activity coefficient. Consequently, in this classical picture, either a decrease in interfacial energy or an increase in the solute concentrations will increase the nucleation rate, which is then exponential in the ratio of γ^3/s^2 . (Note that, while this statement is strictly correct, to compare two systems one cannot completely ignore the kinetic pre-factor, which multiplies the exponential of the free energy barrier. This pre-factor depends on kinetic barriers associated with atomistic processes like ion desolvation and binding and cannot be related to the free energy barrier.)

Now consider what happens as supersaturation is increased in the CaCO_3 system. For simplicity, let's assume that we keep the absolute concentrations of CaCl_2 and NaHCO_3 equal. Then as the solute concentration is increased from zero, no nucleation will occur in either the bulk solution or on the proteins until we cross the solubility limit of the most stable phase, which is calcite. Now one might think that, because the supersaturation with respect to calcite is the same regardless of whether nucleation happens on the template or in the solution, as long as the interfacial energy is lowered by the template, nucleation will happen first on the template. However, the volume fraction of the template matters, because the nucleation probability is proportional to the number of potential nucleation sites, so a miniscule amount of the protein template in a large volume of solution may still be ineffective in controlling nucleation, even if the interfacial energy is very low on the

template. This then sets the *lower limit of protein concentration* for the template to be effective. If self-assembly into larger structures is either required or more effective, then the lower limit of protein concentration is higher still — however, there is no upper limit. Moreover, as a practical matter, the supersaturation has to reach some minimum value for nucleation to be seen on a laboratory timescale, even with an effective template. This, then sets the *lower limit of solute concentration* for the template to be effective.

To understand the upper limit of solute concentration and the effect of having multiple polymorphs on these limits requires some further consideration. If the solute concentration is further increased until the solubility limit of aragonite and vaterite are crossed, then there is a finite probability that any one of the three crystal phases will nucleate, both in solution and on the template. Here again, one might think that a template designed for, say, vaterite will then always cause vaterite to nucleate rather than aragonite or calcite. However, because calcite is less soluble than aragonite and aragonite less so than vaterite, the supersaturation with respect to calcite is always larger than that for aragonite, which is, in turn, larger than that for vaterite. Consequently, in order to predict which polymorph will nucleate requires a knowledge of the ratio of g^3/s^2 (again ignoring the kinetic pre-factor in the expression for the nucleation rate). Obviously, even if the template is highly favorable towards vaterite, if the solute concentration barely exceeds the solubility limit for vaterite, the template will not be effective. Consequently, the lower limit of solute concentration at which the template is effective depends on *both* the value of the interfacial energy relative to that in the bulk solution *and* its value relative to that for the other polymorphs.

The upper limit of solute concentration for the template to be effective arises from two sources. First, at some point the solubility of ACC is reached and, because it has a very low interfacial energy — being a hydrated, disordered phase — it will nucleate in solution at very low supersaturation. It may even nucleate on the template first, but not due to any structural match, rather just due to chemical compatibility. This is not an interesting regime and does not allow us to test out the ability to design templates for crystal polymorphs. *It is for this reason we chose a solute concentration well below the solubility limit of ACC.* The second source arises from a common feature of polymorphic systems: the less stable the phase, the lower the interfacial energy. Consequently, as the solute concentration is increased beyond the solubility limit of each polymorph, there comes a point at which the ratio of g^3/s^2 flips in favor of the less stable polymorph. This is believed to be the reason that, in the absence of protein, one commonly observes the formation of vaterite in bulk solution before any other phase when the solute concentration is sufficiently high (but below the solubility limit of ACC). (This phenomenon is referred to as Ostwald's rule of stages.) By using a solute concentration at which all of these phases can form *and* at which vaterite forms first in bulk solution, one can then test whether a protein template is truly effective at polymorph selection. *Knowing from the literature and our own experience that solutions containing 5 mM CaCl_2 and 5 mM NaHCO_3 places us in this regime, we chose these concentrations to test the effectiveness of our templates.*

Despite this understanding rooted in classical nucleation theory, we do not feel it is wise to say how the process works *in vivo*. There are just too many unknowns about natural systems. Moreover, we know with certainty that numerous organisms utilize ACC as a precursor and that there is essentially no aqueous phase available. Recent findings even point to the use of dense liquid phases created through liquid-liquid phase separation and stabilized by highly charged proteins. While templates can still be important for determining the polymorph that appears, its crystallographic orientation and the facets that are stabilized, this regime is too far from that explored in our study for us to make meaningful extrapolations.

With regards to the role of self-assembly on setting the lower limit for the protein to be effective, we can say that we observe nucleation first next to the Ca-protein supramolecular assemblies, so they are more effective individually when compared to a monomeric protein. However, because they dissolve away as the crystals nucleate they represent a small number of nucleation sites compared to the monomeric proteins and so are, in fact, less effective.

To address this comment, we have made two changes to the manuscript:

- 1) On page 4 we modified the text to read, “We added 1 μM of each designed protein or a BSA control to a mineralization solution containing 5 mM CaCl_2 and 5 mM NaHCO_3 . At this concentration, the solution is supersaturated with respect to all three common crystal phases—vaterite, aragonite, and calcite, but is undersaturated with respect to amorphous CaCO_3 (ACC). Moreover, vaterite, which is the least stable phase, is the first to nucleate in bulk solution. Thus, we are able to test out the ability of the protein templates to nucleate specific polymorphs and alter the nucleation pathway.
- 2) On page 5 we added, “Thus, while these assemblies are individually more effective templates than are the protein monomers, as demonstrated by the rapidity with which nucleation occurs in their vicinity, by the time the nucleation phase has ended, the fraction of the crystals created by the monomers far exceeds that produced by these assemblies.”

Supplementary information:

Where the supplementary Figures were referred to in the main text there are comments above, but additional comments are listed below. For some reason the referencing style is different in the supplementary information (X) numbers rather than superscript numbers. Also it would be good to add page numbers to the supplementary to make it easier to navigate.

Response: Thanks for this comment. We have modified the reference to ensure consistency between SI and main text, and also added the page numbers in the SI. Additional changes may be made during copy editing.

Materials and Methods.

You used many more different chemicals and products than are listed in the materials section. Please complete this.

Response: We have added the list of chemicals used at the beginning of the materials section.

At the end of the designed protein flattening protocol section, there is an extra space between protocol and and, "...RosettaScript XML files implementing the protocol and the resulting models..."

Response: Corrected.

Sometimes you have a space between the numbers and the units, sometimes you don't. For example "...1mM PMSF, 1 mg/mL DNase, pH 8..." should have a space between 1 and mM. This varies throughout and should be corrected in all cases.

Response: Thanks for notifying us. We have carefully checked them and revised them.

Please check your units are consistent with "per" for example, for using mg/mL or mg mL⁻¹.

Response: Corrected.

The TEV cleavage buffer recipe is in size 12 rather than size 11 font, check your font size and type are consistent.

Response: Corrected.

Specify if the % proportions in the buffer recipes are v/v w/v or w/w.

Response: The proportions in the buffer recipes are complete now.

24,000 xg – needs the x in it and the space

Response: Corrected.

dialyzed should be spelled in US English to be consistent with the rest of the manuscript.

Response: Corrected

How long was each of the 5 L dialysis against TBS carried out for, and at what temperature – room temperature or 4 °C? Also the same for the final dialysis into 10 mM MOPS buffer – you say

3 times overnight, so is that 3x 24 hours, or were you in the lab changing over the buffer solutions every 4 hours through the night? Or are you lucky enough to have a machine to do this for you?

Response: Each dialysis was carried out for 2 hours at room temperature and the final overnight cleavage was carried out at 4 °C. The final dialysis into 10 mM MOPS was carried out 3 times, changing the buffer every 24 hours at 4 °C. Thanks for pointing these out for clarification.

The following highlighted text was added:

“Dialysis was carried out three times overnight **at 4 °C (i.e. changing the buffer every 24 hours)** with a dialysis ratio of 1:10000 volume each time.”

Check your capitalizations are correct: cleavage, pET21b, imidazole, CHAPS,

Response: Corrected.

Explain your abbreviations the first time you use them, e.g. IMAC, CHAPS, MOPS, etc. You have done this with some and not with others.

Response: These abbreviations have been corrected, thanks for pointing this out.

Cryo-TEM – “...lacey carbon grid is surface plasma...” should be ...was surface plasma... double check you have got all your tenses right throughout the work.

Response: You are right! We changed the tenses problem in the new version.

Did you do the cryo-TEM with different concentrations of proteins – what were these concentrations? Also, elsewhere you specify different concentrations of calcium and bicarbonate that you say were used in Cryo-TEM samples – why? Or correct it if it's not correct. The temperature of the liquid ethane should be below what temperature? I don't know what you mean by “...throughout the holder.”

Response: We did not perform cryo-TEM at different protein concentrations. The final mineralization solution in both the cryo-TEM and ex situ/in situ experiments comprised 5 mM CaCl₂, 1.08 μM protein, and 5 mM NaHCO₃.

The temperature of the liquid ethane was about -183 °C, which was low enough to vitrify the water. The temperature of the liquid nitrogen was -196 °C, which was low enough to maintain the sample in a vitrified state. The phrase “throughout the holder” is not accurate. To address this comment, we made the following changes to the main text (original SI):

“Using an automated vitrification robot (FEI Vitrobot Mark III, blot time: 3 s), a 3.0 μl sample extracted from a mineralization solution containing equal volumes of 10 mM CaCl₂, 2.16 μM protein, and 10 mM NaHCO₃ was loaded onto a grid and plunged into liquid ethane at about -183 °C. The vitrified sample was saved and transferred in a liquid nitrogen cooled Gatan cryogenic holder operating near -196

°C. The particle morphology and size were determined on a FEI ETEM operated at 300 kV.”

LP-TEM. Change the comma to a full stop and add a . These, “...0 × 200 μm² windows for imaging, were plasma cleaned...” How were they leak checked? How long did the leak check take?

Response: To ensure TEM safety, we utilized a high-vacuum pumping station from Hummingbird Scientific Company to perform leak checking prior to loading the holder into the TEM. The station features short pumping and venting times, a low base pressure (<1e-6 mbar), and a glass viewing port to allow optical microscopy imaging of the holder tip. The integrated stereo-microscope allows researchers to inspect and test the seal of a liquid cell assembly before loading it into the TEM, crucial for protecting vacuum quality. In our case, we typically leak checked the holder in less than 2 mins to ensure we were able to observe the early stage of the nucleation process once the holder was inserted into the TEM.

To address this comment, we changed the description in the main text (original SI) to read:
“The sealed chips were assembled inside the liquid cell holder (Hummingbird Scientific) and were leak-checked for ≈ 2 mins by inserting the assembled holder into a high-vacuum leak checking station (Hummingbird Scientific) which features short pumping and venting times, a low base pressure (<1 × 10⁻⁶ mbar), and a glass viewing port to allow optical microscopy imaging of the holder tip and the liquid-cell.”

Supplementary text

Number density estimation:

Check all your CaCO₃ etc. have the subscript. How you have written the very small numbers is not very clear. It would be better to express 1.85E17 nm³ as 1.85 × 10¹⁷ nm³.

Response: Corrected notations to x 10^N style

Proposed mechanisms of protein control over calcite nucleation:

2nd paragraph begins with a space.

Spaces are wrong – 2 between matches and (202) and no space between (202) and (Fig. S9). “...better matches (202)(Fig. S9).”

Response: Corrected.

Fig S2. (d) LP-TEM image shows ≈ 5 nm CaCO₃

Response: Corrected.

Wherever you discuss the faces, you use the circular parentheses which indicates a specific face. For example (202) is a face, {202} are (202)-like faces (e.g. (022), and (202) are in the same family in calcite, as are some of their negative intergers). Double check you have the notation correct for the face vs the faces in the main text and the supplementary. I noticed this in Fig. S7, but it happens consistently throughout the manuscript. [xxx] for a specific zone axis, and <xxx> for a family of directions, but I don't think this one crops up. You've done it right on the annotations on the TEM images (I think, as mentioned above the resolution is not good enough to read the tiny writing well, but they should probably be the {xxx} ones on the images, and possibly the specific (xxx) ones on the SAED/FFT's if the angles and everything was checked and matched properly).

Response: Thanks for this great comment. We agree that the mark of facets should be more accurate: {} is used to describe a facet group, () is to indicate a specific facet, [] represents a specific zone axis, and <> is to show a growth direction. To resolve this problem, we have carefully checked them and uploaded new versions.

Reviewer #2 (Remarks to the Author):

Designed proteins (DHRs) with flat surfaces displaying functional groups that lattice matched to calcite was used to modulate mineralization. The structural regularity and tunability promote polymorphic specific CaCO₃ nucleation. The authors design a variety of proteins to demonstrate the specificity of their system to drive biomineralization with rational genetic modular design. The manuscript demonstrates that spacing of the carboxyl sidechains can be used to drive different crystal formation. This is a well written manuscript and should be accepted for publication. The results are noteworthy and provides an approach to control surface chemistry to drive biomineralization.

Response: Many thanks for your recognition of this work.

Reviewers' Comments:

Reviewer #1:

Remarks to the Author:

The authors addressed all questions and queries thoroughly and the manuscript looks to be in great shape.

REVIEWERS' COMMENTS

Reviewer #1 (Remarks to the Author):

The authors addressed all questions and queries thoroughly and the manuscript looks to be in great shape.

Response: Thanks for reviewer's recognition for our work!